# Rational $Q$-Systems at Root of Unity
# I. Closed Chains

Jue Hou[1], Yunfeng Jiang[1], Yuan Miao[2]

[1]*School of physics & Shing-Tung Yau Center, Southeast University,*
*Nanjing 211189, P. R. China*
[2] *Kavli Institute for the Physics and Mathematics of the Universe (WPI), The University of Tokyo*
*Institutes for Advanced Study, The University of Tokyo, Kashiwa, Chiba 277-8583, Japan*

### Abstract

The solution of Bethe ansatz equations for XXZ spin chain with the parameter $q$ being a root of unity is infamously subtle. In this work, we develop the rational $Q$-system for this case, which offers a systematic way to find all physical solutions of the Bethe ansatz equations at root of unity. The construction contains two parts. In the first part, we impose additional constraints to the rational $Q$-system. These constraints eliminate the so-called Fabricius–McCoy (FM) string solutions, yielding all primitive solutions. In the second part, we give a simple procedure to construct the descendant tower of any given primitive state. The primitive solutions together with their descendant towers constitute the *complete* Hilbert space. We test our proposal by extensive numerical checks and apply it to compute the torus partition function of the 6-vertex model at root of unity.

# 1 Introduction

Rational $Q$-system [1] provides a powerful way to obtain physical solutions of Bethe equations. The key feature of this method is that it leads to all and only physical solutions. Therefore rational $Q$-system is intimately related to the completeness of Bethe ansatz. Up to now, rational $Q$-system has been constructed for various spin chain models including the XXX [1,2] and XXZ spin chains [3] and their higher rank generalizations [4,5] with periodic,

twisted [6] and some open [3, 7] boundary conditions. Generalization to spin-$s$ case was achieved very recently [8].

The rational $Q$-systems of XXX and XXZ models have significant differences. For XXX spin chain, solutions of $Q$-system give the Bethe roots of the highest weight states of SU(2) symmetry. Descendant states are taken into account at a later stage by adding Bethe roots at infinity, related to the spin lowering operator of the global SU(2) symmetry. Therefore for fixed length $L$ and magnon number $M$, the number of solutions for XXX spin chain is $\binom{L}{M} - \binom{L}{M-1}$, which is less than the dimension of the Hilbert space in the $M$-magnon sector $\binom{L}{M}$. For XXZ spin chain with generic $q$ parameter, $i.e.$ not at root of unity, the SU(2) symmetry is broken down to U(1). The number of solutions simply equals $\binom{L}{M}$. Due to this reason, it is generally harder to solve the rational $Q$-system of the XXZ spin chain.

There is a situation which sits in between of XXX spin chain and XXZ at generic $q$ and is (in)famously subtle. This is the XXZ spin chain with $q$ at root of unity. The reason is that although SU(2) symmetry is still broken, there is an enhancement of symmetry. As a result, descendant states with respect to a different type of symmetry emerges, leading to exponential degeneracies of the transfer matrices. So far the exact structure of this symmetry is not fully understood. Proposals include an $\mathfrak{sl}_2$ loop algebra [9–11] and Onsager algebra [12,13]. The existence of such descendant states is reflected in the solution of Bethe ansatz equations. There exist two special kinds of solutions, which are roots at infinity [14] and the so-called Fabricius–McCoy (FM) strings [15,16]. The latter are bound states (Bethe strings) without string deviation that are also related to the root of unity value. Adding such Bethe roots to any *primitive states*, a notion which will be introduced below, does not change the value of the energy and all higher (quasi)-local charges. Therefore these states can be regarded as the descendant states of the primitive state, sharing the same eigenvalue of the transfer matrices up to a possible minus sign.

The existence of FM-strings poses new challenge for solving Bethe ansatz equations, or its reformulations such as $TQ$-relation and rational $Q$-system. In fact, if we naively apply rational $Q$-system constructed for generic $q$ and simply take $q$ at root of unity, we find that the $Q$-system has infinitely many solutions. The reason is that the center of FM-strings cannot be fixed by rational $Q$-system and any value of the center gives a solution. Constructing rational $Q$-system for XXZ Bethe ansatz equations at root of unity is the goal of the current work. To this end, we take the following strategy. We first impose additional constraints to eliminate the FM-strings. We will call the resulting $Q$-system the *constrained* rational $Q$-system. We tested extensively that the constrained $Q$-system gives all the physical *primitive* solutions required by completeness of Bethe ansatz. There are three types of primitive solutions: 1) Solutions without roots at infinity; 2) Solutions with finite roots and roots at $+\infty$; 3) Solutions with finite roots and roots at $-\infty$. Notice that primitive solutions cannot

have roots at both positive and negative infinities. The three types of constrained $Q$-systems are constructed separately.

Once all the primitive solutions are obtained, the rest of solutions can be constructed by adding FM-strings and infinity pairs, *i.e.* roots of the form $\{+\infty^{n_\infty}, -\infty^{n_\infty}\}$. However, due to the intricacy of the underlying symmetry algebra, constructing all the descendant states require some extra non-trivial work. To start with, the possible number of roots at infinity $n_\infty$ are highly constrained and we need to distinguish for which quantum numbers we can add such solutions. Second, for given length $L$ and magnon number $M$, we need to determine the center of the FM-strings. The location of the center of the FM-strings can be fixed via the fusion relations of transfer matrices [12, 17]. Despite that such fusion relations [18, 19] are well-studied, to the best of our knowledge, the fact that the locations of the FM-strings are hidden in the zeros of the higher-spin transfer matrix has not been reported in the literature. Finally, we need to determine how many different ways we can add the FM-strings and infinity pairs, which can be summarized nicely in terms of a Hasse diagram. The primitive state with all its descendants form a descendant tower. The Hilbert space consists of all such descendant towers.

The rest of the paper is structured as follows. In section 2, we introduce the XXZ spin chain at root of unity and discuss its integrability. In section 3, we discuss the structure of Hilbert space with emphasis on FM-strings and roots at infinities. In section 4, we construct the constrained rational $Q$-system which gives all primitive states. The construction of the descendant towers are discussed in section 5. As an application of our method, we compute the torus partition function of the 6-vertex model at root of unity in section 6. We conclude in section 7 and discuss future directions. Technical details and necessary backgrounds are summarized in the appendices. Of special interest is that in appendix C we give a simple algorithm for computing the number of primitive states for given quantum numbers.

## 2    XXZ spin chain at root of unity

The model that we will consider is the quantum XXZ spin-1/2 chain with the anisotropy parameter $\Delta$. The Hamiltonian of a length $L$ spin chain is given by

$$\mathbf{H}_{\text{XXZ}} = \sum_{j=1}^{L-1} \sigma_j^+ \sigma_{j+1}^- + \sigma_j^- \sigma_{j+1}^+ + \frac{\Delta}{2}(\sigma_j^z \sigma_{j+1}^z - 1) + \kappa \sigma_L^+ \sigma_1^- + \kappa^{-1} \sigma_L^- \sigma_1^+ + \frac{\Delta}{2}(\sigma_L^z \sigma_1^z - 1), \quad (2.1)$$

where $\sigma_j^\alpha$ are Pauli matrices acting on the $j$-th site. Here we consider the twisted boundary condition with the twist $\kappa = e^{\mathrm{i}\phi}$. For $\phi = 0$, we recover the periodic boundary condition.

We parametrize the anisotropy parameter $\Delta$ as

$$\Delta = \frac{q + q^{-1}}{2} = \cosh \eta, \qquad q = e^{\eta}. \tag{2.2}$$

The parameter $q$ is the q-deformation parameter from the isotropic XXX model. When the parameter $q$ satisfies the following relations,

$$\eta = \mathrm{i}\pi \frac{\ell_1}{\ell_2}, \qquad q^{\ell_2} = \varepsilon = \pm 1, \tag{2.3}$$

where $\ell_1$ and $\ell_2$ are two co-prime integers, we say that the q-deformation parameter $q$ (hence the anisotropy parameter $\Delta$) is at root of unity value. We are particularly interested in understanding the physical properties of the XXZ spin chain at root of unity. The intricacy of the XXZ spin chain at root of unity is due to the special properties of the representation theory of the quantum enveloping algebra $\mathcal{U}_q(\mathfrak{sl}_2)$, which plays a crucial role in the integrability of the model [20].

## 2.1 Integrability of XXZ spin chain

The XXZ spin chain is integrable [21, 22] and can be solved by algebraic Bethe ansatz (ABA) [23, 24]. In the ABA framework, the Lax operator of the XXZ spin chain reads

$$
\begin{aligned}
\mathbf{L}_{aj}(u) = {} & \sinh(u)\,\frac{\mathbf{K}_a + \mathbf{K}_a^{-1}}{2} + \cosh(u)\,\frac{\mathbf{K}_a - \mathbf{K}_a^{-1}}{2}\,\sigma_j^z + \sinh(\eta)\big(\mathbf{S}_a^+ \sigma_j^- + \mathbf{S}_a^- \sigma_j^+\big) \\
= {} & \frac{1}{2} \begin{pmatrix} e^u\,\mathbf{K}_a - e^{-u}\,\mathbf{K}_a^{-1} & 2\sinh(\eta)\,\mathbf{S}_a^- \\ 2\sinh(\eta)\,\mathbf{S}_a^+ & e^u\,\mathbf{K}_a^{-1} - e^{-u}\,\mathbf{K}_a \end{pmatrix}_j,
\end{aligned} \tag{2.4}
$$

where the operators acting on the auxiliary space $a$ are in the highest-weight irreducible representation of $\mathcal{U}_q(\mathfrak{sl}_2)$,

$$\mathbf{K}_a\,\mathbf{S}_a^\pm\,\mathbf{K}_a^{-1} = q^{\pm 1}\,\mathbf{S}_a^\pm, \quad \big[\mathbf{S}_a^+, \mathbf{S}_a^-\big] = \frac{\mathbf{K}_a^2 - \mathbf{K}_a^{-2}}{q - q^{-1}}. \tag{2.5}$$

One remark is that in (2.4) when writing the Lax operator in the matrix form, the matrix elements are operators in the *auxiliary space*. This is slightly different from the usual way of writing the Lax operator where matrix elements are operators acting on site-$j$ of the spin chain. At the moment we do not specify the representation of the auxiliary space. The Lax operator (2.5) satisfies the $RLL$-relation

$$\mathbf{R}_{ab}(u - v)\mathbf{L}_{aj}(u)\mathbf{L}_{bj}(v) = \mathbf{L}_{bj}(v)\mathbf{L}_{aj}(u)\mathbf{R}_{ab}(u - v), \quad \forall u, v \in \mathbb{C}, \tag{2.6}$$

where $\mathbf{R}_{ab}$ is the quantum $R$-matrix, which satisfies the Yang-Baxter equation. An explicit form of $\mathbf{R}_{ab}$ can be found in [12]. We define the monodromy matrix as follows,

$$\mathbf{M}_a(u, \phi, \{\xi_j\}) = \left( \prod_{j=L}^{1} \mathbf{L}_{aj}(u, \xi_j) \right) \mathbf{E}_a(\phi), \tag{2.7}$$

with the inhomogeneities $\{\xi_j\}$ and twist operator $\mathbf{E}_a(\phi) = \mathrm{diag}(\cdots, \exp(2\mathrm{i}\phi), \exp(\mathrm{i}\phi), 1) = \mathrm{diag}(\cdots, \kappa^2, \kappa, 1)$. In this paper, we only consider the homogeneous case, *i.e.* all $\xi_j = 0$.

Transfer matrix is defined by taking trace of the monodromy matrix in the auxiliary space[1]

$$\mathbf{T}_a(u, \phi) = \mathrm{tr}_a \mathbf{M}_a(u, \phi). \tag{2.8}$$

From the $RLL$-relation (2.6), we can prove that the transfer matrices are in involution,

$$[\mathbf{T}_a(u, \phi), \mathbf{T}_b(v, \phi)] = 0, \quad \forall u, v \in \mathbb{C}. \tag{2.9}$$

This establishes integrability of the model.

## 2.2 Transfer matrices

The above discussions are valid for general auxiliary spaces. In this subsection, we introduce a number of transfer matrices which are important for later discussions. For more details of these representations, we refer to appendices of [12].

**Spin-$s$ representations**  The first important class of transfer matrices correspond to taking the auxiliary space to be the spin-$s$ $(2s \in \mathbb{Z}_+)$ highest weight representation of $\mathcal{U}_q(\mathfrak{sl}_2)$. We will denote these transfer matrices by $\mathbf{T}_s(u, \phi)$. The special case $s = 1/2$ corresponds to the transfer matrix of the 6-vertex model. The transfer matrices $\mathbf{T}_s(u, \phi)$ will play a crucial role in the fusion relations that will be introduced shortly.

**$\ell_2$-dimensional representations**  At root of unity $q = \exp(\mathrm{i}\ell_1 \pi/\ell_2)$, another important class of transfer matrices correspond to taking the auxliary space to be the $\ell_2$-dimensional highest weight representations. These transfer matrices, which will be denoted by $\mathbf{T}^{\mathrm{hw}}(u, s)$ where $s$ is the complex spin, has been crucial to derive the quasi-local charges of the XXZ spin chain at root of unity [25–28] and to construct the Baxter's $Q$-operator [12].

---

[1]One should not be confused by the index $a$ on the transfer matrix. After taking the trace in auxiliary space, the transfer matrix is an operator acting on the quantum space. However, for different choices of auxiliary space, the explicit form of the transfer matrix can be different. Here the label $a$ is simply used to highlight this fact.

A remarkable property of $\mathbf{T}^{\text{hw}}(u, s)$ is that it can be factorized into two operators [12]

$$\mathbf{T}^{\text{hw}}(u, s) = \mathbf{Q}(x)\mathbf{P}(y), \quad u, s \in \mathbf{C},$$
$$x = u + \frac{2s+1}{2}\eta, \quad y = u - \frac{2s+1}{2}\eta, \tag{2.10}$$

where $\mathbf{Q}(x)$ and $\mathbf{P}(y)$ can be diagonalized simultaneously by on-shell Bethe states.

The $\mathbf{Q}$-operator satisfies the following $TQ$-relation [22]

$$\mathbf{T}_{1/2}(u)\mathbf{Q}(u) = \kappa^{1/2}T_0(u + \eta/2)\mathbf{Q}(u - \eta) + \kappa^{-1/2}T_0(u - \eta/2)\mathbf{Q}(u + \eta), \tag{2.11}$$

where

$$T_0(u) = \sinh(u)^L. \tag{2.12}$$

Therefore $\mathbf{Q}(u)$ is identified with Baxter's $Q$-operator. For an $M$-magnon state, the eigenvalue of $\mathbf{Q}(u)$ is a Laurent polynomial of $t = e^u$ of order $M$

$$Q(u) \propto \prod_{m=1}^{M} \sinh(u - u_m) \propto \prod_{m=1}^{M} \left(t - t^{-1}t_m^2\right). \tag{2.13}$$

Taking the limit $u \to u_m$ in (2.11), we obtain the Bethe ansatz equations

$$\left(\frac{\sinh(u_j + \eta/2)}{\sinh(u_j - \eta/2)}\right)^L = \kappa^{-1} \prod_{k \neq j}^{M} \frac{\sinh(u_j - u_k + \eta)}{\sinh(u_j - u_k - \eta)}. \tag{2.14}$$

We label the eigenstates of the transfer matrices by the zeros of the eigenvalues of the $\mathbf{Q}$-operator, *i.e.* Bethe roots

$$|\{u_m\}_{m=1}^{M}\rangle. \tag{2.15}$$

These eigenstates can be constructed by the algebraic Bethe ansatz (ABA),

$$|\{u_m\}_{m=1}^{M}\rangle \propto \prod_{m=1}^{M} \mathbf{B}(u_m)| \Uparrow\rangle, \tag{2.16}$$

where $\mathbf{B}(u)$ is the B-operator in ABA [24], and the vacuum $| \Uparrow\rangle$ is the ferromagnetic state with all spin-ups.

Now we turn to the operator $\mathbf{P}(y)$ whose eigenvalue will be denoted by $P(u)$. For an $M$-magnon state, $P(u)$ is a Laurent polynomial in $t$ of order $(L - M)$. The zeros of $P(u)$

corresponds to the Bethe roots of the dual state obtained by flipping all spins simultaneously

$$|\{v'_m\}_{m=1}^{L-M}\rangle \propto \prod_{j=1}^{L} \sigma_j^x |\{u_m\}\rangle_{m=1}^{M}. \tag{2.17}$$

Similarly, for an $M$-magnon eigenstate, the eigenvalue of $\mathbf{P}(u)$ is a Laurent polynomial of $t = e^u$ of order $(L - M)$

$$
\begin{aligned}
\mathbf{P}(y)|\{u_m\}\rangle_{m=1}^{M} &= P(u)|\{u_m\}\rangle_{m=1}^{M}, \\
P(u) &\propto \prod_{m'=1}^{L-M} \sinh(u - v'_m) \propto \prod_{m=1}^{L-M} \left[t - t^{-1}(t'_m)^2\right],
\end{aligned} \tag{2.18}
$$

where $t'_m = \exp(v'_m)$.

In terms of ABA, analogously we have

$$|\{v'_m\}_{m=1}^{L-M}\rangle \propto \prod_{m=1}^{L-M} \mathbf{B}(v'_m)|\Uparrow\rangle, \quad |\{u_m\}_{m=1}^{M}\rangle \propto \prod_{m=1}^{L-M} \mathbf{C}(v'_m)|\Downarrow\rangle, \tag{2.19}$$

where $\mathbf{C}(u)$ is the C-operator in ABA [24], and the vacuum $|\Downarrow\rangle$ is the ferromagnetic state with all spin-downs.

It is thus tempting to identify $P(u)$ with the dual solution of $Q(u)$ of the $TQ$-relation. Indeed these two quantities are intimately related, but with subtle differences depending on the choice of twists. When the twist $\phi$ is not *commensurate* [12], $P(u)$ is indeed identified with the dual solution of $Q(u)$. However, when the twist $\phi$ is *commensurate*, the dual solution of $Q(u)$ becomes a *quasi-polynomial* [3], while $P(u)$ remains a Laurent polynomial of order $(L - M)$. We will explain the relations between the dual solution of $TQ$-relation and the eigenvalues of $\mathbf{P}(u)$ in more details in [29].

# 3 States of XXZ model at root of unity

In this section, we discuss the structure of the Hilbert space of the XXZ spin chain at root of unity. Before doing so, let us first briefly recall the structure for the XXX spin chain and XXZ spin chain with generic $q$. For XXX spin chain with periodic boundary condition, states in the Hilbert space are organized in terms of SU(2) multiplets. Each multiplet consists of a highest weight state, corresponding to an on-shell Bethe state with *finite* Bethe roots, together with its descendants obtained by adding roots at infinity. An $M$-magnon Bethe state and its descendants form a spin-$(\frac{L}{2}-M)$ representation of SU(2) algebra with dimension $(L - 2M + 1)$. All the states in the same multiplet share the same eigenvalue of the transfer

matrix $\mathbf{T}_{1/2}(u)$.

For XXZ spin chain with generic $q$, SU(2) symmetry is broken into a U(1) symmetry. For each $M$-magnon Bethe state with finite Bethe roots, in general there is only one state that is degenerate and share the same eigenvalue of $\mathbf{T}_{1/2}(u)$. This is the state with all spins flipped simultaneously.

At root of unity, the situation is more complicated. The degeneracy of $\mathbf{T}_{1/2}(u)$ is generally larger than 2, which signifies an enhanced symmetry. However, since SU(2) invariance is broken, this symmetry cannot be SU(2), but has to be something else. The precise nature of this new symmetry is not fully understood. Conjectures include hidden Onsager algebraic symmetry of the model [12, 13] and the $\mathfrak{sl}_2$-loop algebra [9–11]. In the XXX case, degenerate states are obtained by adding roots at infinity. Similarly, at root of unity, degenerate states can also be obtained by adding special types of roots to a given solution. The difference is that, now there are two types of special roots we can add : 1) Fabricius–McCoy strings; 2) Infinity pairs. Here by 'infinity pairs', we mean the roots with equal number of positive and negative infinities, $i.e.$ $\{+\infty^{n_\infty}, -\infty^{n_\infty}\}$. One remark is that different from the XXX spin chain where the sign of infinities are not relevant, for the XXZ case it is necessary to distinguish the sign of the infinities because they lead to different analytic behaviors of the $Q$-function.

It is straightforward to show that by adding these two types of special roots to a given solution, the eigenvalue of the transfer matrix $\mathbf{T}_{1/2}(u)$ can at most change a sign. This leads us to the notion of a *descendant tower*. A descendant tower is defined as all the states that are degenerate with respect to the eigenvalues of the transfer matrix $\mathbf{T}_{1/2}(u)$ up to a possible minus sign. The *primitive* state is the one with smallest number of Bethe roots. The existence of the descendant towers is closely related to the twist $\kappa$, as explained in [12]. For a generic twist $\kappa$, where neither the FM-string nor the infinity pairs are allowed, the Hilbert space has the same structure as the generic $q$ case.

We would like to emphasize that, unlike the XXX spin chain, primitive solutions can contain roots at infinity. However, in this case it can contain only positive or negative infinities, but not both. In other words, we have the following three types of primitive solutions

1. Finite Bethe roots

$$\{u_1, \ldots, u_N\}, \qquad -\infty < u_k < \infty \qquad (3.1)$$

2. Solutions with positive infinities

$$\{u_1, \ldots, u_N, \underbrace{\infty, \ldots, \infty}_{n_+}\}, \qquad -\infty < u_k < \infty \tag{3.2}$$

3. Solutions with negative infinities

$$\{u_1, \ldots, u_N, \underbrace{-\infty, \ldots, -\infty}_{n_-}\}, \qquad -\infty < u_k < \infty \tag{3.3}$$

In the rest of this section, we give more detailed discussions of the FM-string and roots at infinities.

The $Q$-function of a generic solution of Bethe ansatz equations has the following decomposition

$$Q(t) = Q_{\text{reg}}(t)Q_{+\infty}(t)Q_{-\infty}(t)Q_{\text{FM}}(t). \tag{3.4}$$

The zeros of the regular part

$$Q_{\text{reg}}(t) = \prod_{m=1}^{n_{\text{reg}}} \left(t - t^{-1}t_m^2\right) \tag{3.5}$$

consists of finite Bethe roots $u_m = \log t_m$ that are not part of FM-strings.

The other parts are

$$Q_{+\infty}(t) = t^{-n_+}, \quad Q_{-\infty}(t) = t^{n_-}, \quad Q_{\text{FM}}(t) = \prod_{m=1}^{n_{\text{FM}}} \left(t^{\ell_2} - t^{-\ell_2}t_{\text{FM},m}^{2\ell_2}\right), \tag{3.6}$$

where $n_\pm$ are the number of Bethe roots at $\pm\infty$ respectively. Notice that for the Bethe roots at infinities, we need to take proper limit of the $Q$-functions. Specifically, we need the following two limits

$$-\lim_{u_m \to +\infty} \left(t_m^{-2}t - t^{-1}\right) = t^{-1}, \tag{3.7}$$

$$\lim_{u_m \to -\infty} \left(t - t^{-1}t_m^2\right) = t, \tag{3.8}$$

which justify the form of $Q_{\pm\infty}$ in (3.6).

## 3.1   Exact Fabricius–McCoy strings

Fabricius–McCoy strings are a special type of bound states in the XXZ model at root of unity [15, 16]. They are a set of $\ell_2$ Bethe roots of the following form

$$u_n = u_0 + \frac{\mathrm{i}\pi(n-1)}{\ell_2}, \quad u_0 \in \mathbb{C}, \qquad n \in \{1, 2, \cdots \ell_2\}. \tag{3.9}$$

Such bound states have interesting properties. For example, the scattering phase between an FM-string and any other Bethe roots is trivial [14–16], $i.e.$

$$\prod_{n=1}^{\ell_2} S(u, u_n) = 1, \qquad S(u, v) = \frac{\sinh(u - v - \eta)}{\sinh(u - v + \eta)}. \tag{3.10}$$

In addition, they do not carry any conserved charges except for a $\pi$ momentum when $\varepsilon = q^{\ell_2} = -1$. This implies that the states that have the same regular parts but different FM-strings are degenerate with respect to the transfer matrices $\mathbf{T}_s$ up to a possible sign $\varepsilon$ [12]. The FM-strings are also related to the phantom states in the XXZ model at root of unity [30, 31].

Finally, and most relevant to our current discussions, the presence of FM-strings makes it difficult to solve Bethe ansatz equations or $TQ$-relation. Consider the eigenvalue equations of the $TQ$-relation (2.11)

$$T_{1/2}(u)Q(u) = \kappa^{1/2}T_0(u + \eta/2)Q(u - \eta) + \kappa^{-1/2}T_0(u - \eta/2)Q(u + \eta). \tag{3.11}$$

Suppose $Q(u)$ is a solution to this equation. We can construct another solution by

$$\widetilde{Q}(u) = Q(u)Q_{\mathrm{FM}}(u) \tag{3.12}$$

where $Q_{\mathrm{FM}}(u)$ is the Baxter polynomial of an FM-string which has the following property

$$Q_{\mathrm{FM}}(u \pm \eta) = \varepsilon\, Q_{\mathrm{FM}}(u), \qquad \varepsilon = q^{\ell_2} = \pm 1. \tag{3.13}$$

It is then clear that $\widetilde{Q}(u)$ satisfy the following $TQ$-relation

$$\varepsilon\, T_{1/2}(u)\widetilde{Q}(u) = \kappa^{1/2}T_0(u + \eta/2)\widetilde{Q}(u - \eta) + \kappa^{-1/2}T_0(u - \eta/2)\widetilde{Q}(u + \eta). \tag{3.14}$$

When $\varepsilon = 1$, this is the same $TQ$-relation as (3.11). However, when constructing $\widetilde{Q}(u)$ we did not specify the center $u_0$ of the FM-string. This means for any choice of $u_0$, $\widetilde{Q}(u)$ is a solution of the $TQ$-relation. So there are infinitely many solutions to the $TQ$-relation. This fact makes it impossible to find solutions with FM-strings. Similar analysis for Bethe

ansatz equations using the property (3.10) leads to the same problem. This seems to be a contradiction, as the dimension of the Hilbert space is finite. The resolution is that the center of the FM-string $u_0$ is not arbitrary and need to be determined by other means. One way is by diagonalizing the **Q**-operator constructed in [12]. We will give another way to fix the value of $u_0$ in section 5 by using the fusion relations of the transfer matrices $\mathbf{T}_s$.

## 3.2 Bethe roots at infinity

Now we discuss Bethe roots at infinity. When we say a Bethe root is at infinity, we always mean the Bethe root given in the additive variable $u_k$ instead of the multiplicative variable $t_k = e^{u_k}$ [14]. Whether roots at infinities are allowed depends on quantum numbers $L, M$ and the parameters of the model including twist $\kappa$ and anisotropy $q$ [12]. Let us denote the number of positive and negative infinities by $n_+$ and $n_-$ respectively.

To begin with, we notice that the solutions with $n_+ = n_- = 0$ are always possible. For $n_\pm \neq 0$, they have to satisfy several constraints. The first constraints come from Bethe ansatz equations. By taking the limit $u_j \to \pm\infty$, (2.14) become

$$q^{\pm L} = \kappa^{-1} q^{\pm 2(M - n_\pm)}, \tag{3.15}$$

Here we assume that the Bethe roots at either $+\infty$ or $-\infty$ do not scatter with themselves. In addition, we have the constraint that

$$0 \leq n_\pm \leq \ell_2 - 1. \tag{3.16}$$

For more explanations of this constraint, see [12].

Equation (3.15) and the constraint (3.16) is our starting point for analyzing roots at infinities. When $\kappa$ is not an integer power of $q$, the Bethe equation (3.15) does not permit any solution for integer $n_\pm$. This implies that for a generic twist, there is no solution with roots at infinity.

However, when $\kappa$ is an integer power of $q$, there will be solutions with the integers $n_\pm \leq \ell_2 - 1$. Suppose that $\kappa = q^k$ with $k \in \mathbb{Z}$, (3.15) becomes

$$(\pm 2 n_\pm \pm L + k \mp 2M) \mod (x\, \ell_2) = 0, \tag{3.17}$$

where $x = 1$ for even $\ell_1$ and $x = 2$ for odd $\ell_1$.

When $\kappa \neq \pm 1$, the only permitted solutions to (3.15) are

$$n_+ \neq 0, n_- = 0 \qquad \text{or} \qquad n_+ = 0, n_- \neq 0 \tag{3.18}$$

These solutions correspond to *primitive states* in the absence of FM-strings.

Let us focus on the special case when $\kappa = 1$ (*i.e.* periodic boundary condition) with no FM-string. There are three possibilities for the solution to (3.15) with $n_\pm \neq 0$,

$$n_+ \neq 0, n_- = 0 \quad \text{or} \quad n_- \neq 0, n_+ = 0 \quad \text{or} \quad n_+ = n_- \neq 0. \tag{3.19}$$

The first two cases correspond to primitive states, while the third case corresponds to descendant states. The reason is as follows:

For the first two cases with $n_\pm \neq 0$, using the decomposition (3.4) and the $TQ$-relation, we see that the regular part of the $Q$-polynomial is a solution to a *twisted $TQ$-relation* with effective twist $q^{\pm 2n_\pm}$ that originates from roots at infinities,

$$T_{1/2}(u)Q_{\text{reg}}(u) = q^{\pm n_\pm}T_0(u + \eta/2)Q_{\text{reg}}(u - \eta) + q^{\mp n_\pm}T_0(u - \eta/2)Q_{\text{reg}}(u + \eta). \tag{3.20}$$

$Q_{\text{reg}}(u)$ in this case is not a solution to the original $TQ$-relation, due to the presence of the effective twist. Hence the first two cases are primitive and they have descendants with additional FM-strings.

For the third case, it is easy to observe that since $Q_{+\infty}(t)Q_{-\infty}(t) = 1$, the regular part of the $Q$-polynomial corresponds to a primitive state satisfying the *original $TQ$-relation*,

$$T_{1/2}(u)Q_{\text{reg}}(u) = T_0(u + \eta/2)Q_{\text{reg}}(u - \eta) + T_0(u - \eta/2)Q_{\text{reg}}(u + \eta). \tag{3.21}$$

Therefore, the states with infinity pairs $\{(+\infty)^{n_\infty}, (-\infty)^{n_\infty}\}$ $(n_\infty = n_+ = n_-)$ are descendants of the state corresponding to $Q_{\text{reg}}$.

# 4   Primitive states and constrained $Q$-system

In this section, we construct the constrained $Q$-system which gives all primitive solutions of the XXZ Bethe ansatz equations at root of unity. Our strategy is imposing extra constraints to eliminate solutions containing FM-strings.

As discussed in the previous section, we need to consider two situations : 1) Solutions with finite Bethe roots; and 2) Solutions containing roots at infinities. In what follows, we will first review the rational $Q$-system for XXZ spin chain at generic $q$ and explain why this construction cannot be applied directly to the case when $q$ is a root of unity. We will then discuss the constrained $Q$-system for the primitive states with finite Bethe roots and roots at infinity.

## 4.1 Rational $Q$-system for XXZ spin chain

Let us recall the rational $Q$-system for the XXZ spin chain with twisted boundary condition for generic $q$ [1–3]. Each $Q$-system is associated with a Young diagram as in figure 4.1. For XXZ spin chain with length $L$ and magnon number $M$, we consider a two-row Young diagram with number of boxes $(M, L - M)$ where $L - M \geq M$. We define a $Q$-function at each node of the Young diagram labelled by $Q_{a,s}(t)$ where $a$ and $s$ are the lattice coordinates of the node. Here we work with multiplicative variable $t = e^u$.

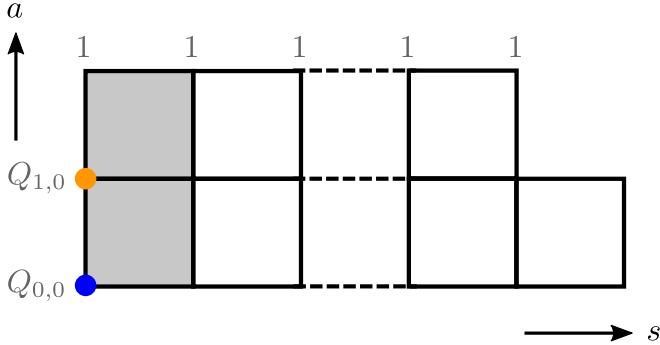

Figure 4.1: Young diagram with two rows.

$QQ$-**relation**    $Q$-functions at different nodes are related by the $QQ$-relation

$$Q_{a+1,s}(t)Q_{a,s+1}(t) = \kappa_a\, Q^-_{a+1,s+1}(t)Q^+_{a,s}(t) - Q^+_{a+1,s+1}(t)Q^-_{a,s}(t)\,, \tag{4.1}$$

where $f^\pm(t) \equiv f(tq^{\pm 1/2})$ and $\kappa_a$ are related to twists. Therefore what is important is the ratio between $\kappa_0$ and $\kappa_1$, which is identified with the twist $\kappa$ in (2.14) as $\kappa = \kappa_0/\kappa_1$.

**Boundary condition**    Certain $Q$-functions at the boundaries of the Young diagram are fixed completely or partially. These are called boundary conditions. More precisely, the upper boundary is completely fixed $Q_{2,s} = 1$ and the $Q$-functions at the left boundary are

$$Q_{0,0}(t) = (t - t^{-1})^L, \qquad Q_{1,0}(t) = \prod_{j=1}^{M}(t - t^{-1}t_j^2) \tag{4.2}$$

where the zeros of $Q_{1,0}(t)$ correspond to Bethe roots, which we want to find by solving the $Q$-system. Since $Q_{1,0}(t)$ is a Laurent polynomial of degree $M$, we can write the following ansatz

$$Q_{1,0}(t) = Q(t) = t^M + \sum_{k=1}^{M-1} c_k\, t^{2k-M} + \frac{t^{-M}}{c_0}\,. \tag{4.3}$$

We will derive a set of algebraic equations for the coefficients $\{c_j\}$, which can be solved and give $Q(t)$. Finding the zeros of $Q(t)$, we obtain the Bethe roots. The algebraic equations come from the zero remainder conditions, which we now turn to.

**Zero remainder condition** We require that all the $Q$ functions on the Young diagram to be Laurent polynomials in $t$, or polynomials in $t$ and $t^{-1}$. From the $QQ$-relation and the boundary condition, we can easy solve $Q_{1,n}(t)$

$$Q_{1,n}(t) = D_{\kappa_1}^n Q(t) \tag{4.4}$$

where $D_\kappa$ is the finite difference operator defined by

$$D_\kappa f(t) = \kappa \, f(tq^{1/2}) - f(tq^{-1/2}) \,. \tag{4.5}$$

From (4.4), it is clear that all $Q_{1,n}(t)$ are Laurent polynomials in $t$. Now we turn to $Q_{0,n}$. From $QQ$-relation, we have

$$Q_{0,n}(t) = \frac{\kappa_0 \, (D_{\kappa_1}^n Q)^+ Q_{0,n-1}^- - (D_{\kappa_1}^n Q)^- Q_{0,n-1}^+}{D_{\kappa_1}^n Q} \,. \tag{4.6}$$

We can use this relation to find $Q_{0,n}$ recursively. Now $Q_{0,n}(t)$ is no longer a Laurent polynomial for generic $Q(t)$. In order for $Q_{0,n}$ to be polynomials, we require that the remainders of the ratio of two polynomials in (4.6) to vanish. Such conditions are called zero remainder conditions, which can be written as a set of algebraic equations for $\{c_j\}$. We then solve this system of algebraic equations and find the solutions.

**Bethe ansatz equations** The Bethe ansatz equations and related $TQ$-relations can be derived from the $QQ$-relation. From the $QQ$-relation of the first row, we find

$$Q_{1,1}^+(t_k) = \kappa_1 \, Q_{1,0}^{++}(t_k), \qquad Q_{1,1}^- = -Q_{1,0}^{--}(t_k) \tag{4.7}$$

while from the second row we have

$$\kappa_0 \, Q_{1,1}^-(t_k)Q_{0,0}^+(t_k) - Q_{1,1}^+(t_k)Q_{0,0}^-(t_k) = 0 \,. \tag{4.8}$$

Combining these two equations and eliminating $Q_{1,1}(t)$, we find

$$\kappa_0 \, Q_{1,0}^{--}(t_k)Q_{0,0}^+(t_k) + \kappa_1 \, Q_{1,0}^{++}(t_k)Q_{0,0}^-(t_k) = 0 \,. \tag{4.9}$$

Plugging in the explicit form of $Q_{1,0}(t)$ and $Q_{0,0}(t)$, we obtain the Bethe ansatz equation

$$\left(\frac{t_j^2 q - 1}{t_j^2 - q}\right)^L = -\frac{\kappa_1}{\kappa_0} \prod_{k=1}^M \frac{t_j^2 q^2 - t_k^2}{t_j^2 - t_k^2 q^2} = \kappa^{-1} \prod_{k=1}^M \frac{t_j^2 q^2 - t_k^2}{t_j^2 - t_k^2 q^2}, \tag{4.10}$$

which is identical to the BAE mentioned previously (2.14) after a change of variables.

**Root of unity**   Let us now discuss why the above construction is problematic at root of unity. The main reason is due to the appearance of FM-strings.

Similar to the $TQ$-relation, suppose we have found a solution of the zero remainder condition $Q(t)$, which means all $Q_{0,n}$ calculated from

$$Q_{0,n} = \frac{\kappa_0 \, (D_{\kappa_1}^n Q)^+ Q_{0,n-1}^- - (D_{\kappa_1}^n Q)^- Q_{0,n-1}^+}{D_{\kappa_1}^n Q} \tag{4.11}$$

are Laurent polynomials of $t$. Now consider the following $Q(t)$ function by multiplying the $Q$-function of an FM-string

$$\widetilde{Q}(t) = Q(t) Q_{\mathrm{FM}}(t). \tag{4.12}$$

Let us first consider the case where $\ell_1$ is even. In this case, we have $Q_{\mathrm{FM}}^+(t) = Q_{\mathrm{FM}}^-(t)$ and

$$D_{\kappa_1}\widetilde{Q} = \kappa_1 \, Q^+ Q_{\mathrm{FM}}^+ - Q^- Q_{\mathrm{FM}}^- = Q_{\mathrm{FM}}^+ \left(\kappa_1 Q^+ - Q^-\right) = Q_{\mathrm{FM}}^+ D_{\kappa_1} Q \tag{4.13}$$

It is then easy to prove that

$$D_{\kappa_1}^n \widetilde{Q} = (Q_{\mathrm{FM}}^+)^n \, D_{\kappa_1}^n Q. \tag{4.14}$$

Therefore

$$\begin{aligned}
Q_{0,n} &= \frac{\kappa_0 \, (D_{\kappa_1}^n \widetilde{Q})^+ Q_{0,n-1}^- - (D_{\kappa_1}^n \widetilde{Q})^- Q_{0,n-1}^+}{D_{\kappa_1}^n \widetilde{Q}} \\
&= \left(\frac{Q_{\mathrm{FM}}}{Q_{\mathrm{FM}}^+}\right)^n \frac{\kappa_0 \, (D_{\kappa_1}^n Q)^+ Q_{0,n-1}^- - (D_{\kappa_1}^n Q)^- Q_{0,n-1}^+}{D_{\kappa_1}^n Q}
\end{aligned} \tag{4.15}$$

For even $\ell_1$, we have $Q_{\mathrm{FM}}^+(t) = \pm Q_{\mathrm{FM}}(t)$, therefore the factor $(Q_{\mathrm{FM}}(t)/Q_{\mathrm{FM}}^+(t))^n = (\pm 1)^n$ is a simple global factor. As a result, $Q_{0,n}$ in (4.15) are polynomials. This implies that $\widetilde{Q}(t)$ is another solution of the zero remainder condition. Since we have not specify the center of the FM-string, it can be a free parameter. This means the zero remainder condition in this case has infinitely many solutions.

The analysis for odd $\ell_1$ is similar. If we consider an even number of FM-strings, the situation is the same as the $\ell_1$ even case. In the case for odd number of FM-strings solutions $Q_{\mathrm{FM}}^+(t) = -Q_{\mathrm{FM}}^-(t)$. The zero-remainder condition will be modified slightly, which effectively corresponds to adding a $\pi$ twist to the system (or a $\pi$ boost for the momentum).

To conclude, due to the presence of FM-strings, the rational $Q$-system will yield infinitely many solutions at root of unity. This is easily confirmed by numerical calculations. Clearly there is a discrepancy between the infinitely many solutions of the rational $Q$-system and the finitely many physical solutions bound by the dimension of the entire Hilbert space. This can be resolved by adding constraints on the $Q$-functions, resulting in the correct number of *physical* primitive solutions, as shown in the next section.

## 4.2   Constraints on $Q$-functions

Since the existence of FM-strings makes it hard to solve $Q$-system, we shall forbid such solutions in the first step and add them back in a later stage. This can be achieved by imposing constraints for the $Q$-functions.

We start with $Q$-functions with only finite Bethe roots. We first notice that if the Bethe roots contain FM-strings, the coefficients of the $Q$-function are not independent, but satisfy specific relations. These are precisely the relations we want to forbid. The $Q$-function is parameterized as

$$Q(t) = t^M + \sum_{k=1}^{M-1} c_k t^{2k-M} + \frac{t^{-M}}{c_0} = \prod_{j=1}^{M}\left(t - \frac{t_j^2}{t}\right). \qquad (4.16)$$

where $t_j$ are Bethe roots. Note that the coefficient for $t^{-M}$ is $1/c_0$. The reason for this unusual choice (instead of the seemingly more natural choice $c_0$) is to ensure the coefficient $1/c_0$ to be non-zero, which is true for $Q$-functions corresponding to finite Bethe roots.

If $Q(t)$ contains an FM-string, it must take the form

$$Q(t) = Q_{\mathrm{FM}}(t)\tilde{Q}(t) \qquad (4.17)$$

where

$$Q_{\mathrm{FM}}(t) = \prod_{j=1}^{\ell_2}\left(t - \frac{(t_j^{\mathrm{FM}})^2}{t}\right) \qquad (4.18)$$

and $\tilde{Q}(t)$ is another Laurent polynomial of degree $(M - \ell_2)$. The FM string center is defined

as $t_j^{\text{FM}} = \exp(u_j^{\text{FM}})$, where $u_j^{\text{FM}}$ satisfies (3.9). From the ansatz (4.16), we have

$$Q(t) = t^M + \sum_{k=1}^{M-1} c_k t^{2k-M} + \frac{t^{-M}}{c_0} = \prod_{j=1}^{\ell_2} \left( t - \frac{(t_j^{\text{FM}})^2}{t} \right) \prod_{k=\ell_2+1}^{M} \left( t - \frac{t_k^2}{t} \right) \tag{4.19}$$

Expanding right hand side of (4.19), we can solve $\{c_j\}$ in terms of $\{t_j^{\text{FM}}\}$ and $\{t_k\}$ ($k = \ell_2 + 1, \ldots, M$) and write

$$c_l = F_l(\{t_j^{\text{FM}}\}, \{t_k\}), \qquad l = 0, 1, \ldots, M - 1. \tag{4.20}$$

The above equation can be written as a set of polynomial equations of three sets of variables $\{c_l\}$, $\{t_j^{\text{FM}}\}$ and $\{t_k\}$. We can follow standard algorithms in elimination theory (see for example Chapter 3 of [32]) to eliminate $\{t_j^{\text{FM}}\}$ and $\{t_k\}$ from (4.20), resulting in a set of equations involving only $\{c_j\}$

$$R_k(c_0, \ldots, c_{M-1}) = 0, \qquad k = 1, \ldots, N_c, \tag{4.21}$$

where $R_k(c_0, \cdots, c_{M-1})$ are polynomials of $\{c_l\}$ and $N_c$ is different from $M$ in general and its explicit value depends on the elimination process. If the solution contains an FM-string, the coefficients of $Q(t)$ satisfy additional relations (4.21). Therefore, to eliminate FM-strings, we can add the following constraint to the rational $Q$-system

$$1 + w \left( |R_1|^2 + \ldots + |R_{N_c}|^2 \right) = 0 \tag{4.22}$$

where $|R_k|^2 = R_k R_k^*$ and $w$ is an extra variable which we introduce to guarantee the constraint $(|R_1|^2 + \ldots + |R_{N_c}|^2)$ is non-zero. The equation (4.22) implies that we cannot have the situation $R_1 = R_2 = \cdots R_{N_c} = 0$. One might worry that the constraints (4.22) might involve the complex conjugate $\{c_j^*\}$ and make it complicated. However, we can in fact prove that all $\{c_j\}$ are real. The details of the proof is given in Appendix A.

**Example 3.1** Let us give an example to illustrate how the above procedure works. To this end, we consider the following quantum numbers

$$L = 8, \quad M = 4, \quad \eta = \frac{\text{i}\pi}{3}. \tag{4.23}$$

For this case, the FM-string is a length-3 bound state consisting of the following 3 Bethe roots

$$u_1^{\text{FM}} = \alpha_{\text{FM}} - \frac{2\pi\text{i}}{3}, \qquad u_2^{\text{FM}} = \alpha_{\text{FM}}, \qquad u_3^{\text{FM}} = \alpha_{\text{FM}} + \frac{2\pi\text{i}}{3} \tag{4.24}$$

Note that the imaginary parts of the Bethe roots are defined mod $\pi$. Plugging (4.24) into (4.19), we find that the solution

$$c_0 = \frac{e^{-6\alpha_{\mathrm{FM}}}}{t_4^2}, \qquad c_1 = -e^{6\alpha_{\mathrm{FM}}}, \qquad c_2 = 0, \qquad c_3 = -t_4^2. \tag{4.25}$$

Eliminating $\alpha_{\mathrm{FM}}$ and $t_4$ from (4.25), we find the following relations among $\{c_j\}$[2]

$$c_2 = 0, \qquad c_0 c_1 c_3 - 1 = 0. \tag{4.26}$$

Therefore to obtain the constrained $Q$-system, we add the following equation

$$1 + w\left[c_2^2 + (c_0 c_1 c_3 - 1)^2\right] = 0, \tag{4.27}$$

to the zero remainder conditions.

## 4.3 Primitive solutions with roots at infinity

Now we consider primitive states with roots at infinity. The strategy is the same, but the ansatz is slightly different.

**Positive infinity** For $n_+ \neq 0$, $n_- = 0$, the $Q$-function takes the form

$$Q(t) = \frac{t^{-M}}{c_0} + \sum_{k=1}^{M-n_+-1} c_k t^{2k-M} + t^{M-2n_+}. \tag{4.28}$$

To find the constraints for the FM-string, we solve $\{c_k\}$ from

$$Q(t) = \frac{t^{-M}}{c_0} + \sum_{k=1}^{M-n_+-1} c_k t^{2k-M} + t^{M-2n_+} = \frac{1}{t^{n_+}} \prod_{j=1}^{\ell_2} \left(t - \frac{(t_j^{\mathrm{FM}})^2}{t}\right) \prod_{k=\ell_2+1}^{M-n_+} \left(t - \frac{t_k^2}{t}\right) \tag{4.29}$$

and eliminate $\{t_j^{\mathrm{FM}}\}$ and $\{t_k\}$ from the solution.

**Negative infinity** For $n_+ = 0, n_- \neq 0$, the $Q$-function takes the form

$$Q(t) = t^M + \sum_{k=n_-}^{M-1} c_k t^{2k-M} \tag{4.30}$$

---

[2]Such computations can be performed straightforwardly by computer algebraic systems, such as the function `Eliminate` of Wolfram Mathematica.

To find the constraints for the FM-string, we solve $\{c_k\}$ from

$$Q(t) = t^M + \sum_{k=n_-}^{M-1} c_k t^{2k-M} = t^{n_-} \prod_{j=1}^{\ell_2} \left( t - \frac{(t_j^{\mathrm{FM}})^2}{t} \right) \prod_{k=\ell_2+1}^{M-n_-} \left( t - \frac{t_k^2}{t} \right) \tag{4.31}$$

and eliminate $\{t_j^{\mathrm{FM}}\}$ and $\{t_k\}$ from the solution.

**Example 3.2** Here we present an example with root at infinity. We consider the following quantum numbers

$$L = 10, \qquad M = 5, \qquad \eta = \frac{\mathrm{i}\pi}{3}, \qquad n_+ = 1. \tag{4.32}$$

The Bethe roots of the FM-string are still (4.24). Plugging into (4.28), we find that the solution is given by

$$c_0 = \frac{e^{-6\alpha_{\mathrm{FM}}}}{t_4^2}, \qquad c_1 = -e^{6\alpha_{\mathrm{FM}}}, \qquad c_2 = 0, \qquad c_3 = -t_4^2. \tag{4.33}$$

Eliminating $\alpha_{\mathrm{FM}}$ and $t_4$ from (4.25), we find the following relations among $\{c_j\}$

$$c_2 = 0, \qquad c_0 c_1 c_3 = 1. \tag{4.34}$$

Therefore to obtain the constrained $Q$-system, we add the following equation

$$1 + w \left[ c_2^2 + (c_0 c_1 c_3 - 1)^2 \right] = 0, \tag{4.35}$$

to the zero remainder conditions.

# 5 Descendant states

In this section, we discuss the construction of descendant states of a given primitive state. Combined together, they form a multiplet which we shall call a *descendant tower*. States within each descendant tower share the same regular part of the $Q$-functions, by definition.

## 5.1 General discussions

**Descendant tower** Let us denote the Bethe roots of the primitive state by $\{u_m\}$, $m = 1, \dots, M$ with $M \le L/2$. The Bethe roots of the descendant states are given by adding *infinity pairs*[3] or FM-strings to $\{u_m\}$, until one reaches the unique 'bottom' state. The bottom

---

[3] Recall that these are the Bethe roots which consists of $n_\infty$ pairs of $+\infty$ and $-\infty$ roots.

state is nothing but the dual state of $|\{u_m\}\rangle$, defined by flipping all spins simultaneously

$$|\{v'_m\}\rangle \equiv \prod_{j=1}^{L} \sigma_j^x |\{u_m\}\rangle,\tag{5.1}$$

where $\{v_{m'}\}$, $m' = 1, 2, \ldots, L - 2M$ are the Bethe roots of the dual state. All the descendant states of a primitive state form a diamond shaped Hasse diagram [33] whose end points are the primitive state and its dual state. Examples of such diagrams can be found in figure 5.2, 5.3, 5.4 of the next subsection.

**FM-strings** Now we discuss how to find all the FM-strings of a given primitive state. We consider a state with length $L$, magnon number $M$, twist $\kappa = e^{i\phi}$ and anisotropy $\eta = i\pi \ell_1 / \ell_2$. Let us define the $Q$-function of the primitive solution

$$Q(u) = \prod_{j=1}^{M} \sinh(u - u_j).\tag{5.2}$$

Our claim is that the FM-strings are given by the zeros of the following quantity

$$F(u) = \sum_{k=0}^{\ell_2 - 1} e^{ik\phi} \frac{\left[\sinh\left(u + (k + \frac{1}{2})\eta\right)\right]^L}{Q(u + k\eta)Q(u + (k + 1)\eta)}.\tag{5.3}$$

In other words, we have

$$F(u) \propto \prod_{j=1}^{n_{\text{FM}}} \left(\prod_{m=1}^{\ell_2} \sinh\left(u - \alpha_j^{\text{FM}} - \frac{2m - 1 - \ell_2}{2\ell_2}i\pi\right)\right)\tag{5.4}$$

where each set of Bethe roots

$$u_j^{(m)} = \alpha_j^{\text{FM}} + \frac{2m - 1 - \ell_2}{2\ell_2}i\pi, \qquad m = 1, \ldots, \ell_2\tag{5.5}$$

form an FM-string with center $\alpha_j^{\text{FM}}$. The total number of the FM-strings is denoted by $n_{\text{FM}}$. For a proof of this claim, see appendix B.

**Infinity pairs** From the definition (5.3), one might expect that $F(u)$ is a trigonometric polynomial in $u$ (or Laurent polynomial in the multiplicative variable) of degree $L - 2M$. However, there are cases when this is not the case, which signifies the appearance of infinity

pairs $\{(-\infty)^{n_\infty}, (+\infty)^{n_\infty}\}$. The number of infinity pairs $n_\infty$ is given by

$$n_\infty = \frac{1}{2}(L - 2M - \deg F). \tag{5.6}$$

**Hasse diagram** The structure of the descendant tower can be seen clearly from a Hasse diagram. It consists of all possible paths from the primitive state to the dual state, at each step one adds an FM-string or infinity pairs.

**The algorithm** To summarize, given a primitive solution $\{u_m\}$, the descendant tower can be constructed by the following procedure

1. Construct the $Q$-polynomial (5.2) and compute the corresponding $F(u)$ in (5.3);

2. Compute the number of infinity pairs by (5.6);

3. Determine the center of FM-strings $\{\alpha_j^{\text{FM}}\}$ by finding the zeros of $F(u)$;

4. Construct the Hasse diagram of the descendant tower by adding either infinity pairs or an FM-string once at a time, until one reaches the dual state.

By using the solutions to the constraint $Q$-systems and the algorithm to construct the descendant towers, we are able to find all the physical solutions to the XXZ spin chain at root of unity. Those structures are essential to compute the torus partition functions of the 6-vertex model, cf. Section 6, and they reveal certain intriguing combinatorics as shown in Appendix C.

## 5.2 Examples

To see how the outlined procedure works in practice, let us consider a few examples in this subsection. It is more convenient to work with multiplicative variables. With slight abuse of notation, we denote the $Q$-polynomial in multiplicative variables by

$$Q(t) = \prod_{j=1}^{M} \left( t\, t_j^{-1} - t^{-1} t_j \right) \tag{5.7}$$

The quantity $F(u)$ in (5.3) written in terms of multiplicative variables (omitting an unimportant factors $2^{-L+2M}$) reads

$$F(t) = \sum_{k=0}^{\ell_2 - 1} e^{ik\phi} \frac{\left[ tq^{k+1/2} - 1/(tq^{k+1/2}) \right]^L}{Q(tq^k)Q(tq^{k+1})}. \tag{5.8}$$

**Example 5.1** Consider a spin chain with $L = 6$, $\phi = 0$, $\eta = 2i\pi/3$ ($\ell_1 = 2, \ell_2 = 3$). We consider the $M = 0$ sector, which corresponds to the pseudovacuum state $|\uparrow\uparrow\uparrow\uparrow\uparrow\uparrow\rangle$. Its dual state is the one with all spins down $|\downarrow\downarrow\downarrow\downarrow\downarrow\downarrow\rangle$. The $Q$-polynomial in this case is a constant

$$Q(t) = 1. \tag{5.9}$$

Plugging in (5.8), we obtain

$$F(t) = 3(t^6 - 20 + t^{-6}) \tag{5.10}$$

We find that $\deg F = 6 = L$, therefore we do not have infinity pairs. Since each FM-string is of length-3 in this case, we conclude we have 2 FM-strings. Denoting their centers by $t_1^{\mathrm{FM}}$ and $t_2^{\mathrm{FM}}$, we have

$$F(t) \propto (t^6 - 20 + t^{-6}) = \prod_{j=1}^{2}(t/t_j^{\mathrm{FM}} - t_j^{\mathrm{FM}}/t)(qt/t_j^{\mathrm{FM}} - t_j^{\mathrm{FM}}/qt)(t/qt_j^{\mathrm{FM}} - qt_j^{\mathrm{FM}}/t), \tag{5.11}$$

from which we find that

$$t_1^{\mathrm{FM}} = \left(10 - 3\sqrt{11}\right)^{1/6}, \qquad t_2^{\mathrm{FM}} = \left(10 + 3\sqrt{11}\right)^{1/6}. \tag{5.12}$$

The Hasse diagram for the descendant tower is given by figure 5.2. This descendant tower contains 4 states.

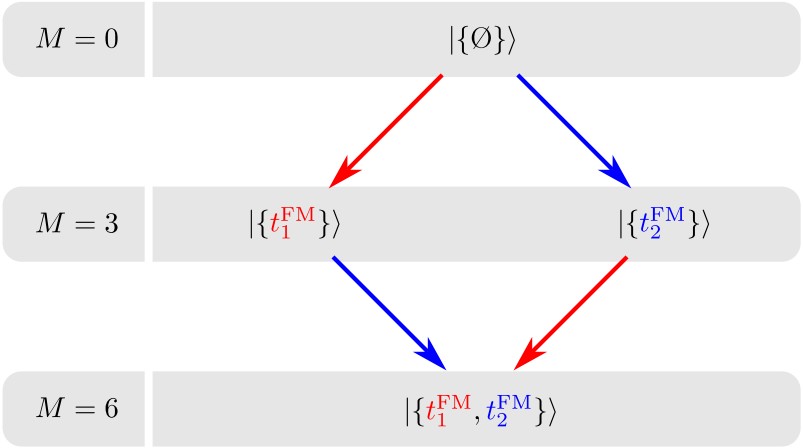

Figure 5.2: Hasse diagram for the descendant tower of **Example 5.1**.

**Example 5.2** Consider a spin chain with $L = 8$, $\phi = 0$, $\eta = 2i\pi/3$ ($\ell_1 = 2, \ell_2 = 3$).

We consider the one magnon sector $M = 1$ with $t_1 = \mathrm{i}$. The $Q$-polynomial is given by

$$Q(t) = t + t^{-1}. \tag{5.13}$$

The corresponding $F(t)$ reads

$$F(t) = 3(t^6 - 29 + t^{-6}) \tag{5.14}$$

Since $\deg F = 6 = L - 2M$, we do not have infinity pairs. Since $\ell_2 = 3$, each FM-string is of length 3. Therefore we have 2 FM-strings. Similar to **Example 5.1**, we can find the center of the FM-strings

$$t_1^{\mathrm{FM}} = \left(\frac{1}{2}(29 + 3\sqrt{93})\right)^{1/6}, \qquad t_2^{\mathrm{FM}} = \left(\frac{1}{2}(29 - 3\sqrt{93})\right)^{1/6}. \tag{5.15}$$

The Hasse diagram for the descendant tower is given in figure 5.3. This descendant tower contains 4 states.

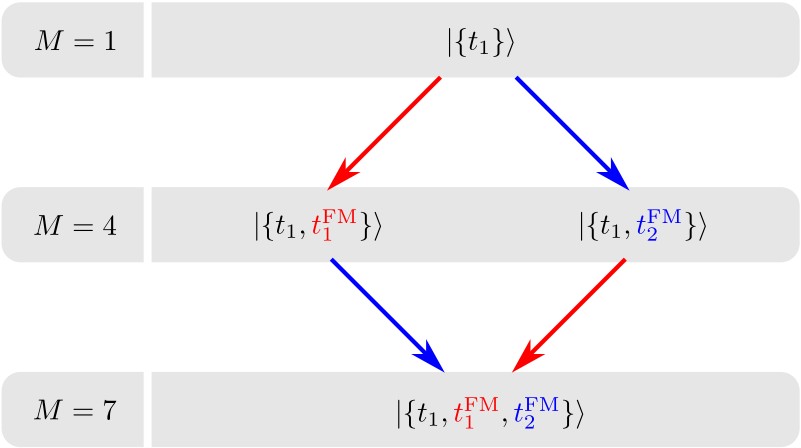

Figure 5.3: Hasse diagram for the descendant tower of **Example 5.2**.

**Example 5.3** Let us consider an example with infinity pairs. We take $L = 12$, $\phi = 0$, $\eta = \mathrm{i}\pi/3$ ($\ell_1 = 1, \ell_2 = 3$). We consider the one-magnon state with

$$t_1 = \left(\frac{\sqrt{3} + 1}{\sqrt{3} - 1}\right)^{1/2}, \qquad Q(t) = \sqrt{2 - \sqrt{3}}\, t - \frac{\sqrt{2 + \sqrt{3}}}{t}. \tag{5.16}$$

The corresponding $F(t)$ is given by

$$F(t) = -9\left(2(8 + 3\sqrt{3})t^6 + 59 + 2(8 - 3\sqrt{3})t^{-6}\right) \tag{5.17}$$

We see that in this case $\deg F = 6 < L - 2M = 10$. Therefore the number of infinity pairs is $n_\infty = 2$. The Hasse diagram for the descendant tower is given in figure 5.4. This descendant tower contains 8 states.

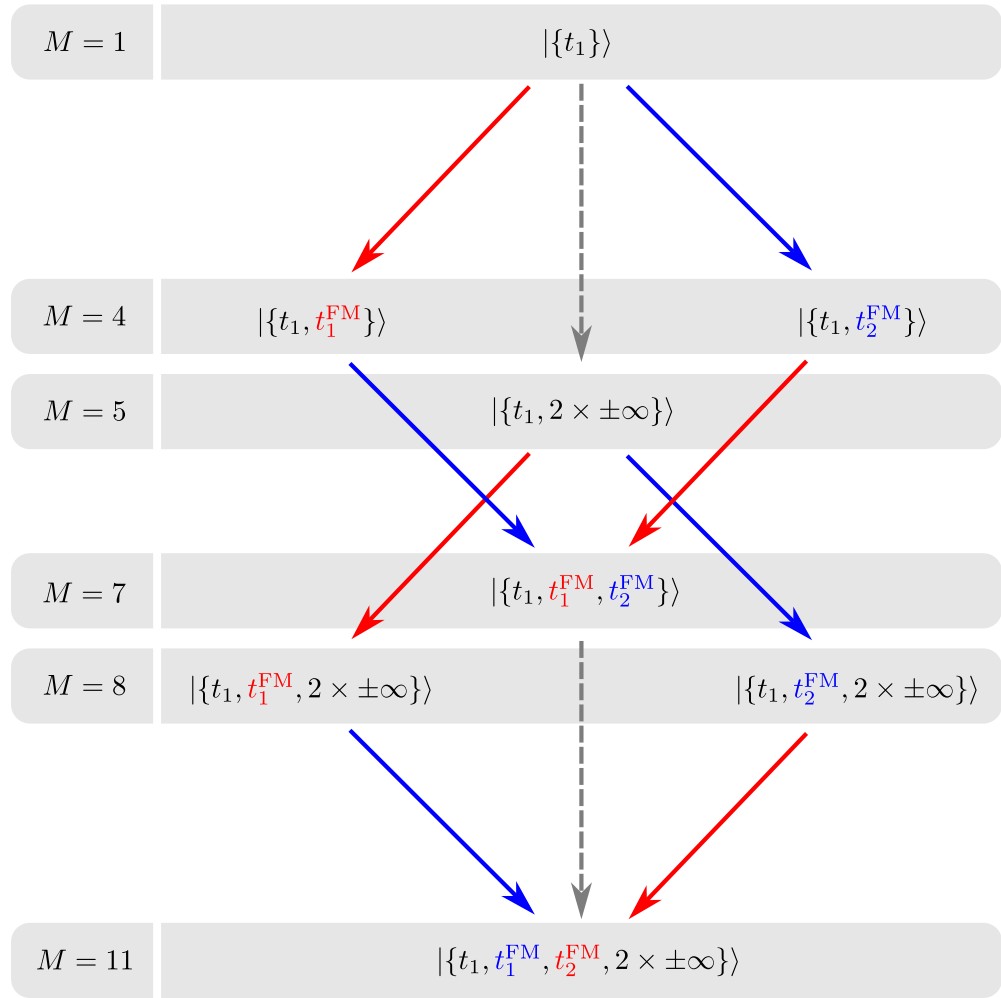

Figure 5.4: Hasse diagram for the descendant tower of **Example 5.3**. $\pm\infty$ should be understood as the value of the Bethe roots. In terms of the variable $t = e^u$, we have $\lim_{u \to +\infty} t = +\infty$ and $\lim_{u \to -\infty} t = 0$.

**Example 5.4** $L = 8$, $\eta = \mathrm{i}\pi/3$, $\phi = 0$. We would like to demonstrate the mirroring structure of the descendant towers when the primitive state possesses either $+\infty$ or $-\infty$ root(s) but not pairs of them. We focus on a primitive state with one $+\infty$ root and one regular root ($M = 2$ and $n_+ = 1$):

$$Q_1(t) = t^{-2} - \frac{1}{2}(\sqrt{3} + 1) \propto t^{-1}(t - t^{-1}t_1^2). \tag{5.18}$$

The regular root $t_1$ is

$$t_1 = \sqrt{\sqrt{3} - 1}. \tag{5.19}$$

Because of the $\mathbb{Z}_2$ symmetry ($\prod_n \sigma_n^x$) of the model, the transfer matrix eigenvalues are degenerate between the primitive state $|\psi\rangle$ and its partner state "beyond the equator" $\prod_n \sigma_n^x |\psi\rangle$. In this example, the state "beyond the equator" contains the same regular root $t_1$, an FM-string and $\ell_2 - 1 = 2$ roots at $-\infty$ [12]. This hints us to consider another primitive state with $(\ell_2 - 1) = 2$ roots at $-\infty$ and the same regular root (5.19)

$$Q_2(t) = t - \frac{1}{2}(\sqrt{3} + 1)t^3 \propto t^2(t - t^{-1}t_1^2). \tag{5.20}$$

We can calculate the corresponding $F(t)$, which is the same for both primitive states up to a normalization factor,

$$
\begin{aligned}
F_1(t) &\propto t^3 \left[ \sqrt{\frac{31 - 15\sqrt{3}}{22}} t^3 - \sqrt{\frac{22}{31 - 15\sqrt{3}}} t^{-3} \right], \\
F_2(t) &\propto t^{-3} \left[ \sqrt{\frac{31 - 15\sqrt{3}}{22}} t^3 - \sqrt{\frac{22}{31 - 15\sqrt{3}}} t^{-3} \right],
\end{aligned}
\tag{5.21}
$$

where the FM string centre is located at

$$t_1^{\text{FM}} = i \left( \frac{31 + 15\sqrt{3}}{13} \right)^{1/6}. \tag{5.22}$$

There is a subtlety on understanding the function $F(t)$, which is related to the state beyond the equator (the dual state) with respect to the primitive state. For instance,

$$F_1(t) = \frac{Q_1^{\text{dual}}(t)}{Q_1(t)} = t^3 Q_{\text{FM}}(t), \tag{5.23}$$

where the $Q$-function associated with the dual state is

$$Q_1^{\text{dual}}(t) = t^2(t - t^{-1}t_1^2)Q_{\text{FM}}(t). \tag{5.24}$$

This is reflected by the fact that the dual state is the descendant state of the mirrored tower in Figure 5.5

The descendant tower in this case has a mirroring structure: It splits into two parts with the same regular root(s) and FM-strings, while possessing with different number of Bethe roots at $+\infty$ and $-\infty$, respectively. The total number in the mirroring descendant tower is

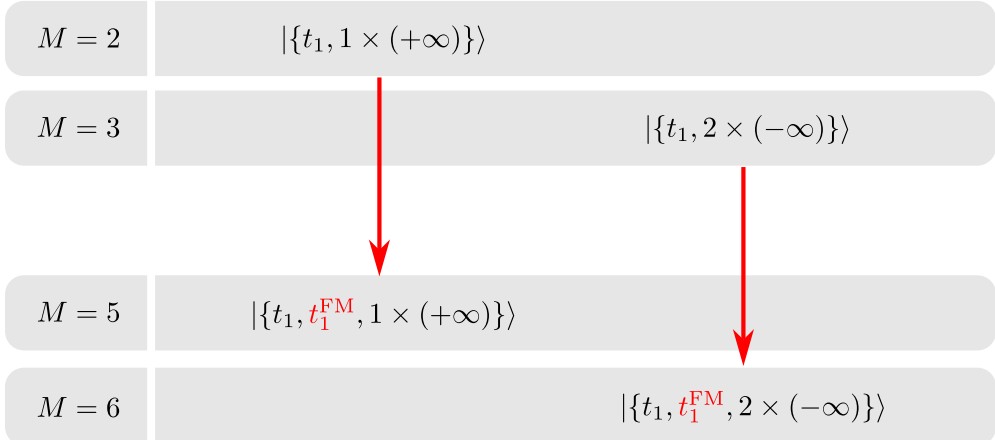

Figure 5.5: Hasse diagram for the descendant tower of **Example 5.4**. $\pm\infty$ should be understood as the value of the Bethe roots, the same as previously explained. It is clear that the dual states of the two primitive states are in the mirrored descendant towers, respectively.

hence $2^{n_{\mathrm{FM}}+1}$.

# 6 Torus Partition Function of 6-vertex model

With the rational $Q$-system we are able to correctly find all the primitive solutions as well as their descendants. This is very useful in many applications. In this section, we apply our method to compute the torus partition functions of 6-vertex model at root of unity on a $N \times M$ lattice, as is shown in figure 6.6.

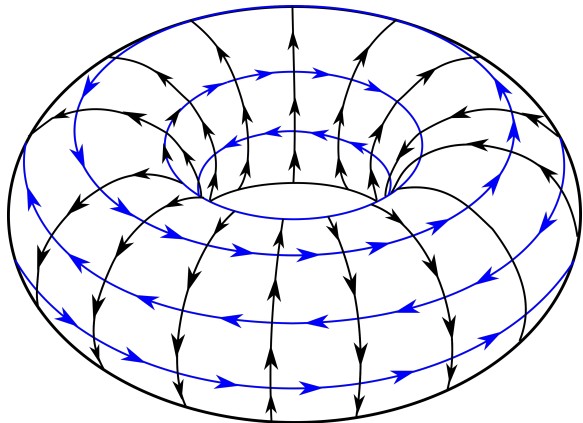

Figure 6.6: Torus partition function of 6-vertex model.

Using the transfer matrix approach, the torus partition function is defined as

$$W_{N,M}(t) = \mathrm{Tr}\big(\mathbf{T}_{1/2}^M(t)\big), \tag{6.1}$$

where $\mathbf{T}_{1/2}(t)$ is the transfer matrix with $N$ sites. The torus partition function $W_{N,M}(t)$ is a Laurent polynomial in $t$ of order $(NM)$ Alternatively, one can rotate the lattice by $\pi/2$ and obtain the same partition function from the other channel

$$W_{M,N}(t) = W_{N,M}(t) = \mathrm{Tr}\big(\tilde{\mathbf{T}}_{1/2}^N(t)\big), \tag{6.2}$$

where $\tilde{\mathbf{T}}_{1/2}(t)$ is the transfer matrix with $M$ sites. This symmetry can be used as a consistency check of our results.

We first solve the $Q$-system to obtain all the primitive $Q$-functions, as well as the descendants (the degeneracies are crucial here). The eigenvalues of the transfer matrix $\mathbf{T}_{1/2}(t)$ can be found by $TQ$-relation

$$T_{1/2}(t) = \frac{1}{Q}\big(T_0^+ Q^{--} + \kappa T_0^- Q^{++}\big). \tag{6.3}$$

For the descendant states, we observe that

$$T_{1/2}^{\mathrm{des}}(t) = \varepsilon^{n_{\mathrm{FM}}} T_{1/2}^{\mathrm{pri}}(t), \tag{6.4}$$

where $T_{1/2}^{\mathrm{pri}}$ and $T_{1/2}^{\mathrm{des}}$ stand for the eigenvalues of the transfer matrix for the primitive and its descendant states respectively.

The advantage of using the $Q$-systems to calculate the torus partition functions is that we would not have to diagonalize a $2^L \times 2^L$ transfer matrix. Instead, we solve the $Q$-system, which are polynomial equations. The number of primitive solutions of $M$ magnons are typically much smaller than the dimension of the Hilbert space for the $M$-magnon sector. This is clearly a significant improvement when the system size $L$ becomes bigger.

We exemplify the calculation for the 6-vertex model with $N = 8$ and $M = 6$, $\Delta = -1/2$, $\phi = 0$. In this case, we have 144 primitive states (28 out of 144 are primitive states with root(s) at infinities) and 112 descendant states, as described in details in table 1.

More explicitly, the torus partition function with $N = 8$ and $M = 6$ is

$$W_{8,6} = \frac{1}{140737488355328} \sum_{m=1}^{24} c_m\big(t^{2m} + t^{-2m}\big) + c_0, \tag{6.5}$$

where the exact coefficients are curated in Appendix D.

For the other channel with $L = 6$, $\Delta = -1/2$, $\phi = 0$. We have 40 primitive states (4 out of 40 are primitive states with root(s) at infinity) and 24 descendant states, as shown explicitly in table 2.

| M | total # | # primitive states | | | # descendant states | | |
|---|---|---|---|---|---|---|---|
| | | $n_\pm = 0$ | $n_+ > 0$ | $n_- > 0$ | $n_+ = n_-$ | $n_+ > 0$ | $n_- > 0$ |
| 0 | 1 | 1 | 0 | 0 | 0 | 0 | 0 |
| 1 | 8 | 8 | 0 | 0 | 0 | 0 | 0 |
| 2 | 28 | 13 | 7 | 7 | 1 | 0 | 0 |
| 3 | 56 | 40 | 7 | 7 | $1 \times 2$ | 0 | 0 |
| 4 | 70 | 54 | 0 | 0 | $8 \times 2$ | 0 | 0 |
| 5 | 56 | 0 | 0 | 0 | 40 | 7 | 7 |
| 6 | 28 | 0 | 0 | 0 | 13 | 7 | 7 |
| 7 | 8 | 0 | 0 | 0 | 8 | 0 | 0 |
| 8 | 1 | 0 | 0 | 0 | 1 | 0 | 0 |

Table 1: The number of primitive states and their descendants for $L = 8$, $\Delta = -1/2$, $\phi = 0$, organized by the number of magnons of each state $M$. The number of primitive states is $\mathcal{N}^{\mathrm{pri}} = 144$, while the number of descendant states is $\mathcal{N}^{\mathrm{des}} = 112$.

| M | total # | # primitive states | | | # descendant states | | |
|---|---|---|---|---|---|---|---|
| | | $n_\pm = 0$ | $n_+ > 0$ | $n_- > 0$ | $n_+ = n_-$ | $n_+ > 0$ | $n_- > 0$ |
| 0 | 1 | 1 | 0 | 0 | 0 | 0 | 0 |
| 1 | 6 | 4 | 1 | 1 | 0 | 0 | 0 |
| 2 | 15 | 13 | 1 | 1 | 0 | 0 | 0 |
| 3 | 20 | 18 | 0 | 0 | $1 \times 2$ | 0 | 0 |
| 4 | 15 | 0 | 0 | 0 | 13 | 1 | 1 |
| 5 | 6 | 0 | 0 | 0 | 4 | 1 | 1 |
| 6 | 1 | 0 | 0 | 0 | 1 | 0 | 0 |

Table 2: The number of primitive states and their descendants for $L = 6$, $\Delta = -1/2$, $\phi = 0$, organized by the number of magnons of each state $M$. The number of primitive states is $\mathcal{N}^{\mathrm{pri}} = 34$, while the number of descendant states is $\mathcal{N}^{\mathrm{des}} = 30$.

Without much surprise, we arrive at

$$W_{6,8} = \frac{1}{140737488355328} \sum_{m=1}^{24} c_m \left( t^{2m} + t^{-2m} \right) + c_0 = W_{8,6}. \tag{6.6}$$

This simply implies the torus partition function remains the same after a $\pi/2$ rotation.

# 7   Conclusions and discussions

In this paper, we address the question of finding all physical solutions of the XXZ spin chain with parameter $q$ being a root of unity. Our method is a refined version of the rational $Q$-system, which has been successfully constructed for various spin chain models [1,3,4,7,8]. A naive application of the rational $Q$-system for XXZ spin chain with generic $q$ fails at root

of unity, due to the appearance of FM-strings. The main new results of this work are: 1) A method which eliminate solutions with FM-strings, rendering only primitive solutions; 2) A systematic method to construct the descendant tower of a given primitive state. With these new developments, we now have an efficient method of obtaining all physical solutions of XXZ spin chain at root of unity.

Even though our method is straightforward to implement, the structure of the solutions is rather intricate. For a given length $L$ and magnon number $M$, whether there are primitive solutions with root(s) at infinities is sensitive to the parameters $\kappa$ and $q$ and need to be considered separately. In addition, for a given primitive solution, the number of FM-strings and infinite pairs are not immediately clear and need to be worked out with some effort. These of course reflect the intricacy of the enhanced symmetry at root of unity. Therefore it would be interesting to further clarify the structure of Bethe solutions with respect to the representation theory of the enhanced symmetry, such as the conjectured Onsager symmetry [13]. In this regard, it is useful to investigate the open chain with $U_q(\mathfrak{sl}_2)$ quantum group symmetry [20, 34, 35], where the representation theory of the $U_q(\mathfrak{sl}_2)$ algebra will be of vital importance. Our $Q$-system can be generalized to this case, following the lines of [36]. The advantage for the open chain case is that, the representation theory is fully understood and it is a clean starting point for clarifying the representation theoretical meaning of the Bethe solutions. A more detailed analysis will be presented elsewhere [37].

The rational $Q$-system constructed in this work has a number of potential applications. Combining with computational algebraic geometry method, rational $Q$-system can be used to compute the lattice partition function of various lattice models of intermediate size. Through the renowned vertex-face correspondence [38–42], the partition functions of the 8-vertex (and 6-vertex) models can be mapped into the partition functions of the Solid-on-Solid (SOS) model, an archetypical example of the interaction-round-a-face (IRF) model [41, 43]. It would be useful if one can apply the rational $Q$-system method to the 8-vertex and SOS models too, extending the current paradigm to the $Q$-systems without the U(1) symmetry.

Another intriguing direction, which was initiated in the work [20] is further exploring the relation with the representation theory of affine Temperley-Lieb (aTL) algebra and the physical solution space of Bethe ansatz equations. It is particularly interesting at root of unity, where rich structure emerges. The representation theory of the aTL algebra has been studied in [20, 44, 45] and it would be very interesting to discover such structures from the solutions of rational $Q$-system.

Finally, it is also interesting to generalize the constrained $Q$-system in this work to higher rank [5] and higher spin models [8] at root of unity, where more $Q$-functions are involved and the nested Bethe ansatz plays a crucial role.

# Acknowledgement

The work of J.H. is supported by Jiangsu Funding Program for Excellent Postdoctoral Talent. The work of Y.J. is partly supported by Startup Funding no. 3207022217A1 of Southeast University. Y.M.'s work was supported by World Premier International Research Center Initiative (WPI), MEXT, Japan. Y.M. is grateful to Jules Lamers and Vincent Pasquier for collaborations on related topics.

# A  Reality of $\{c_j\}$

In this appendix, we prove that $\{c_j\}$ are real for $|q| = 1$. We start from the $TQ$-relation

$$\tau(t)Q(t) = Q_{0,0}(t\,q^{-1/2})Q(t\,q) + Q_{0,0}(uq^{1/2})Q(t\,q^{-1})\,. \tag{A.1}$$

In the regime $|\Delta| \leq 1$ ($|q| = 1$), we have $q^* = 1/q$. [4]

Let us define

$$\overline{Q}(t) = t^M + \sum_{k=1}^{M-1} c_k^* \, t^{2k-M} + \frac{t^{-M}}{c_0^*} \tag{A.2}$$

Taking complex conjugate of both sides of (A.1)[5]

$$\tau(t)\overline{Q}(t) = Q_{0,0}(tq^{1/2})\overline{Q}(tq^{-1}) + Q_{0,0}(tq^{-1/2})\overline{Q}(tq) \tag{A.3}$$

where we have used the fact that $Q_{0,0}^*(tq^{\pm}) = Q_{0,0}(tq^{\mp})$ and $\tau^*(t) = \tau(t)$, which will be proven shortly. Equations (A.1) and (A.3) indicate that both $Q(t)$ and $\overline{Q}(t)$ are solutions of the $TQ$-relation. Since $TQ$-relation is a second order finite difference equation, it has two independent solutions. However, the other solution is 'beyond the equator' and has higher degree. It cannot be identified with $\overline{Q}(t)$. Therefore we must have $Q(t) \propto \overline{Q}(t)$. Since $Q(t)$ and $\overline{Q}(t)$ are both monic, we then conclude that $Q(t) = \overline{Q}(t)$, which implies that $c_j^* = c_j$.

Now we show that $\tau^*(t) = \tau(t)$, namely the eigenvalues of the transfer matrix are real for real $t$. This amounts to prove that the transfer matrix $\mathbf{T}_{1/2}(t)$ is Hermitian. For XXX spin chain, using the standard representation for the Lax operator

$$L_{an}(u) = \begin{pmatrix} u + \mathrm{i}\,S_n^z & \mathrm{i}\,S_n^- \\ \mathrm{i}\,S_n^+ & u - \mathrm{i}\,S_n^z \end{pmatrix}_a \tag{A.4}$$

---

[4]The following is obviously true when $q$ is at root of unity.

[5]Here $t$ is regarded as a formal variable and complex conjugate does not act on it.

we have (assuming $u$ is real)

$$L^\dagger_{an}(u) = \begin{pmatrix} u - \mathrm{i}\, S^z_n & -\mathrm{i}\, S^+_n \\ -\mathrm{i}\, S^-_n & u + \mathrm{i}\, S^z_n \end{pmatrix}_a = \sigma^y_a L_{an}(u)\sigma^y_a \,, \tag{A.5}$$

where we use the simple fact that

$$\sigma^y \begin{pmatrix} A & B \\ C & D \end{pmatrix} \sigma^y = \begin{pmatrix} D & -C \\ -B & A \end{pmatrix}. \tag{A.6}$$

The transfer matrix for a length $L$ spin chain is defined as

$$\mathbf{T}_{1/2}(u) = \mathrm{tr}_a \left[L_{a1}(u)\dots L_{aL}(u)\right] \tag{A.7}$$

We therefore have

$$\begin{aligned}
\mathbf{T}^\dagger_{1/2}(u) &= \mathrm{tr}_a \left[L^\dagger_{aL}(u)\dots L^\dagger_{a1}(u)\right] \tag{A.8}\\
&= \mathrm{tr}_a \left[\sigma^y_a L_{aL}(u)\sigma^a_y \dots \sigma^a_y L_{a1}(u)\,\sigma^a_y\right]\\
&= \mathrm{tr}_a \left[L_{a1}(u)L_{a2}(u)\dots L_{aL}(u)\right] = \mathbf{T}_{1/2}(u)
\end{aligned}$$

For XXZ spin chain, the Lax operator reads

$$L_{an}(t) = \begin{pmatrix} t\, q^{S^z_n} - t^{-1}q^{-S^z_n} & (q - q^{-1})S^-_n \\ (q - q^{-1})S^+_n & t\, q^{-S^z_n} - t^{-1}q^{S^z_n} \end{pmatrix} \tag{A.9}$$

Once again we have $L^\dagger_{an}(t) = \sigma^y_a\, L_{an}(t)\, \sigma^y_a$ for real $t$. It then follows that $\mathbf{T}^\dagger_{1/2}(t) = \mathbf{T}_{1/2}(t)$.

# B    Location of FM-strings

In this appendix, we give more details for the derivation of the the location of FM-strings. In the main text, we claimed that they are encoded in the zeros of the quantity $F(u)$ defined in (5.3) (or equivalently $F(t)$ in (5.8) in terms of multiplicative variables.) We will justify this claim and derive $F(u)$ explicitly. The derivation is based on results from [12], which we briefly review below.

**Factorization of transfer matrix**    It was proven in [12] that the transfer matrix $\mathbf{T}^{\mathrm{hw}}(t, s)$ factorizes into two parts

$$\mathbf{T}^{\mathrm{hw}}(t, s) = \mathbf{Q}(x)\mathbf{P}(y)\,, \tag{B.1}$$

where

$$x = t\, q^{\frac{2s+1}{2}}, \qquad y = t\, q^{-\frac{2s+1}{2}}, \tag{B.2}$$

and $s$ is a complex number. By construction, the eigenvalue of $\mathbf{T}^{\text{hw}}(t, s)$ is a Laurent polynomial of $t$ of order $L$. When $s = (\ell_2 - 1)/2$ (namely $2s + 1 = \ell_2$), we have

$$\mathbf{T}^{\text{hw}}\left(t, \frac{\ell_2 - 1}{2}\right) = \mathbf{T}_{\frac{\ell_2 - 1}{2}}(t), \tag{B.3}$$

where $\mathbf{T}_s(t)$ is the transfer matrix where the auxiliary space is in the unitary spin-$s$ representation. Here $s$ is an half integer. In this case, the factorization property reads

$$\mathbf{T}_s(u) = \mathbf{Q}\left(u + \frac{2s+1}{2}\eta\right)\mathbf{P}\left(u - \frac{2s+1}{2}\eta\right) \tag{B.4}$$

In terms of the eigenvalues, we have

$$T_s(u) = Q\left(u + \frac{2s+1}{2}\eta\right)P\left(u - \frac{2s+1}{2}\eta\right). \tag{B.5}$$

This equation implies that the zeros of $Q(u)$ and $P(u)$ are contained in the zeros of $T_s(u)$. An important fact for us is that the polynomials $Q(u)$ is the solution of the usual $TQ$-relation. The zeros of $Q(u)$ are the Bethe roots.

**Zeros of $P(u)$**  The zeros of $P(u)$ corresponds to the solution 'beyond the equator'. For a given primitive solution $Q(u)$, the zeros of $P(u)$ contains the zeros of $Q(u)$ as a subset, in addition to that, it contains FM-strings and roots at infinity. (For a detailed discussion, one can find in Sec. 7 of [12].)

As a result, we have

$$P(u) = Q(u)F(u) \tag{B.6}$$

where the zeros of $F(u)$ correspond to FM-strings and roots at infinity.

According to (B.5), for a given primitive solution corresponding to $Q(u)$, if we know $T_s(u)$, we can fix $P(u)$ using (B.5), and $F(u)$ simply follows from (B.6). Therefore, one of the key steps is to find $T_s(u)$ from a given $Q(u)$. This can be achieved by the fusion relation of the transfer matrix.

**Fusion of transfer matrix** It is proven in [18] that the transfer matrices satisfy the following fusion relations

$$\mathbf{T}_{1/2}\left(u \pm \frac{2s+1}{2}\eta\right)\mathbf{T}_s(u) = e^{i\phi}\,T_0\left(u \pm (s+1)\eta\right)\mathbf{T}_{s-1/2}\left(u \mp \frac{\eta}{2}\right) \tag{B.7}$$

$$+ T_0\left(u \pm s\eta\right)\mathbf{T}_{s+1/2}\left(u \pm \frac{\eta}{2}\right),$$

where $T_0(u) = [\sinh(u)]^L$ and the eigenvalue of $\mathbf{T}_{1/2}(u)$ can be worked out by the usual $TQ$-relation. Using these two initial conditions, we can work out the eigenvalue of $\mathbf{T}_s(u)$ recursively. This allows us to obtain the eigenvalues of the higher-spin transfer matrices from the eigenvalues of the spin-1/2 transfer matrix. More explicitly, we have the the following interpolation formula

$$T_s(u) = Q\left(u + \frac{2s+1}{2}\eta\right)Q\left(u - \frac{2s+1}{2}\eta\right) \tag{B.8}$$

$$\times \sum_{k=0}^{2s} e^{ik\phi}\frac{T_0(u + (k-s)\eta)}{Q(u + (k-s-\frac{1}{2})\eta)Q(u + (k-s+\frac{1}{2})\eta)}$$

Combining the above equation with (B.5), we find [12]

$$T_s(u) = Q\left(u + \frac{2s+1}{2}\right)Q\left(u - \frac{2s+1}{2}\right) \tag{B.9}$$

$$\times \sum_{k=0}^{2s} e^{ik\phi}\frac{T_0(u + (k-s)\eta)}{Q(u + (k-s-\frac{1}{2})\eta)Q(u + (k-s+\frac{1}{2})\eta)}$$

$$= Q\left(u + \frac{2s+1}{2}\right)P\left(u - \frac{2s+1}{2}\right)$$

$$= Q\left(u + \frac{2s+1}{2}\right)Q\left(u - \frac{2s+1}{2}\right)F\left(u - \frac{2s+1}{2}\right).$$

Comparing the first and last equality, we obtain

$$F(u) = \sum_{k=0}^{2s} e^{ik\phi}\frac{T_0(u + (k+\frac{1}{2})\eta)}{Q(u + k\eta)Q(u + (k+1)\eta)} = \sum_{k=0}^{2s} e^{ik\phi}\frac{\left[\sinh(u + (k+\frac{1}{2})\eta)\right]^L}{Q(u + k\eta)Q(u + (k+1)\eta)}, \tag{B.10}$$

which is (5.3). This derivation provides the justification of the method to find the zeros of FM-strings in Sec. 5.

# C   Combinatorics for the number of primitive states

In some applications, it is useful to know the number of primitive solutions for given quantum numbers. In this case, since we do not need to know the explicit solutions, there are more efficient methods than solving the constrained $Q$-system. In this appendix, we present such an algorithm, based on a conjecture which we have tested numerically. We focus on the case with no twist, $i.e.$ $\kappa = 1$ and $\eta = \frac{i\pi}{3}$ for simplicity. Generalizations to other parameters can be achieved similarly.

In order to determine the number of primitive states with a fixed system size $L$ and number of down spins $M$, we need to carry out the calculation recursively by subtracting the number of descendant states with number of down spins $m < M$.

Another subtlety is number of primitive states with Bethe roots at infinity. Essentially it amounts to counting the number of primitive solutions with only finite Bethe roots of a twisted system with effective twist $\kappa = q^{\pm n_\pm}$ and effective magnon number $M' = M - n_\pm$.

We shall consider the case with even $L$, so that the descendant states are always present. For $M > \frac{L}{2}$, there is no primitive states, and

$$\mathcal{N}^{\mathrm{des}}(L, M) = \binom{L}{M}, \quad M > \frac{L}{2}. \tag{C.1}$$

So we can focus on finding the number of primitive states for $M \le \frac{L}{2}$.

Before introducing the algorithm, we make the following conjecture for the number of primitive states with infinite Bethe root(s) by observing the numerical results:

$$\mathcal{N}^{\mathrm{pri}}_{\pm\infty}(L, M, n_\pm) = \binom{L}{M - n_\pm} - \sum_{x=0}^{M-n_\pm-1} \binom{L}{x}, \tag{C.2}$$

when $n_\pm = n_+ = n_-$ is a solution to (3.15). When there is no solution to (3.15), $\mathcal{N}^{\mathrm{pri}}_{\pm\infty}(L, M, n_\pm) = 0$. It will be useful in the algorithm to obtain the number of primitive states.

Moreover, for each $(L, M)$, the number of primitive states are

$$\mathcal{N}^{\mathrm{pri}}(L, M) = \binom{L}{M} - \mathcal{N}^{\mathrm{des}}(L, M), \tag{C.3}$$

which are further decomposed into three parts,

$$\mathcal{N}^{\mathrm{pri}}(L, M) = \mathcal{N}^{\mathrm{pri}}_{\mathrm{reg}}(L, M) + \mathcal{N}^{\mathrm{pri}}_{+\infty}(L, M, n_+) + \mathcal{N}^{\mathrm{pri}}_{-\infty}(L, M, n_-). \tag{C.4}$$

The number of descendant states for a given $M$ is determined by the number of primitive

states with $\tilde{M} < M$ as well as the structure of the descendant tower, which is explained in Section 5.

**The algorithm**  The algorithm to determine the number of primitive states is achieved recursively.

1. We start with $M = 0$, i.e. the ferromagnetic vacuum. In this case $\mathcal{N}^{\mathrm{pri}}(L, 0) = 1$. We also know its descendants by constructing the descendant tower.

2. Then we move to $M = 1$. $\mathcal{N}^{\mathrm{des}}(L, 1) = 0$ and $\mathcal{N}^{\mathrm{pri}}(L, 1) = L$. We need to distinguish $\mathcal{N}^{\mathrm{pri}}_{\mathrm{reg}}(L, 1)$ and $\mathcal{N}^{\mathrm{pri}}_{\pm\infty}(L, 1)$ because they have different descendant tower structures, cf. Section 5. From (C.2), we obtain $\mathcal{N}^{\mathrm{pri}}_{\pm\infty}(L, 1)$ and $\mathcal{N}^{\mathrm{pri}}_{\mathrm{reg}}(L, 1) = L - \mathcal{N}^{\mathrm{pri}}_{+\infty}(L, 1) - \mathcal{N}^{\mathrm{pri}}_{-\infty}(L, 1)$. Their descendants are obtained accordingly.

3. For $M = 2$, $\mathcal{N}^{\mathrm{des}}(L, 2)$ is determined by the descendant towers of states with $M < 2$. Therefore, we have $\mathcal{N}^{\mathrm{pri}}(L, 2) = \binom{L}{2} - \mathcal{N}^{\mathrm{des}}(L, 2)$. $\mathcal{N}^{\mathrm{pri}}_{\pm\infty}(L, 2)$ are obtained from (C.2) and $\mathcal{N}^{\mathrm{pri}}_{\mathrm{reg}}(L, 2) = \mathcal{N}^{\mathrm{pri}}(L, 2) - \mathcal{N}^{\mathrm{pri}}_{+\infty}(L, 2) - \mathcal{N}^{\mathrm{pri}}_{-\infty}(L, 2)$. We again construct their descendant towers.

4. For $M \geq 3$, we repeat the same procedure as $M = 2$, increasing $M$ one by one. We terminate the algorithm when $M > \frac{L}{2}$.

From the algorithm above, we computed the number of primitive and descendant states for $L = 6$ and $8$, which are in accord with the results of the $Q$-system calculations as shown in tables 1 and 2.

Another curious observation is that

$$\mathcal{N}^{\mathrm{pri}}_{\mathrm{reg}}(L, L/2) = \mathcal{N}^{\mathrm{pri}}(L, L/2) = 2 \cdot 3^{L/2-1}, \tag{C.5}$$

as shown in table 3.Even though the number of primitive states grows exponentially at the equator, it is still of a vanishing factor when going to the thermodynamic limit $L \to \infty$,

$$\lim_{L \to \infty} \frac{\mathcal{N}^{\mathrm{pri}}(L, L/2)}{\mathcal{N}(L, L/2)} = 0, \tag{C.6}$$

with $\mathcal{N}(L, L/2) = \binom{L}{L/2}$. This implies that we cannot simply ignore the contribution of the descendant states to physical quantities such as correlation functions in the thermodynamic limit, e.g. thermodynamic Bethe ansatz (TBA). A similar discussion on the relation between TBA and the descendant states can be found in [12].

A full-fledged understanding of the combinatorics of primitive states would be achieved through studying carefully of the conjectured Onsager symmetry of the XXZ spin chain with periodic boundary condition, which will be reserved for later investigations.

| $L$ | 6 | 8 | 10 | 12 | 14 |
|---|---|---|---|---|---|
| $\mathcal{N}^{\mathrm{pri}}(L, L/2)$ | 18 | 54 | 162 | 1458 | 4374 |

Table 3: The number of primitive states at the equator $(L, L/2)$ for $L$ even, $\Delta = 1/2$ and $\kappa = 1$.

# D   Coefficients for the torus partition function

$$
\begin{aligned}
&c_0 = 57267357244328424573, \quad c_1 = 54076521230629032240, \\
&c_2 = 45526563458718976224, \quad c_3 = 34160205646639767616, \\
&c_4 = 22830887730248006100, \quad c_5 = 13580296134202058160, \\
&c_6 = 7181465323611005168, \quad c_7 = 3371637072931630992, \\
&c_8 = 1403048821385237139, \quad c_9 = 516445615459374816, \\
&c_{10} = 167746535765622792, \quad c_{11} = 47940464791005984, \\
&c_{12} = 12015558187203100, \quad c_{13} = 2630770706010480, \\
&c_{14} = 501082343957136, \quad c_{15} = 82600192303376, \\
&c_{16} = 11723781340854, \quad c_{17} = 1420831897920, \\
&c_{18} = 145966283304, \quad c_{19} = 12459660480, \\
&c_{20} = 881811720, \quad c_{21} = 48704080, \\
&c_{22} = 2323824, \quad c_{23} = 68160, \quad c_{24} = 2837.
\end{aligned}
\tag{D.1}
$$

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
