# Peer review of "Rational Q-systems at Root of Unity I. Closed Chains"

_SciPost Physics_

## Round 2 · Referee Report · Anonymous (Referee 1) · 2024-2-3

Strengths

  1. Has a breakthrough result in a longstanding open problem.
  2. The developed method could be applied in many related topics.
  3. It is very nicely and clearly written.

Weaknesses

  1. As I list in my report, the manuscript could be a little improved.

Report

Integrable spin chains are usually solved by the Bethe ansatz method. The number of solutions of the Bethe ansatz equations are not in one to one correspondence, however, with the eigenvalues of the transfer matrix and the Hamiltonian and it is a difficult case by case investigation to select the correct ones. The rational Q-system, which formulates the QQ equations can overcome this difficulty in the XXX spin chain and also for the XXZ spin chain if q is not a root of unity. The root of unity case is notoriously difficult as Fabricius-McCoy (FM) strings with arbitrary center as well as roots at infinities can appear.

The authors could managed to overcome this difficulty by adapting the Q-system to the root of unity case. The main trick was to eliminate the FM strings and infinite roots by extra constraints in the Q-system and figuring out the way how to add them back, which can be organised in terms of Hesse diagrams. The fusion hierarchy was also crucial in fixing the center of the FM strings. In order to test the method they performed various checks as well as calculated the torus partition function and checked its monodromy invariance.

I think the manuscript is a very nice piece of work, which fills a gap in the literature and will have significant impact on the field that is why I support its publication. I have only a few comments.

The paper is thoroughly written in a pedagogical style, with very clear linear argumentations. Somehow one exception was Example 5.4 for me. I think the logic could be improved a bit there. For instance the author write "In the meantime, we need to consider another primitive state ...". Would it be possible to argue first why do we need this other state at all.

Requested changes

I have also found some typos and some small points.

  1. In the $l_2$-dimensional representations of the transfer matrix it is not clear how the representation depends on the complex spin s. Is it related to $l_2$? Or is it related to the highest weight as for the spin s representations?

  2. The $v$ and $v'$ notation is not systematic in the paper. In (2.18) did the author mean $v$ or $v'$, what is the definition of $t'$? Also consider (5.1).

  3. In (3.7) and (3.8) $u_m$ should be sent to the infinites and not $t_m$-s.

  4. In (3.15) and (3.17) $N$ denotes, what was denoted by $M$ before.

  5. Some articles are not used correctly.

---

## Round 2 · Referee Report · Anonymous (Referee 2) · 2024-2-7

Strengths

1-Tackles interesting problem using novel technqiues 2-Examines the appearance of exclusion of FM-strings in the Q-system approach 3- Works through the proposed approach in a number of simple examples 4-Main result of interest to the community

Weaknesses

1-The statment and proof of the key result lack precision
2-The presentation style sometimes lacks clarity
3-Key references are missing, or not cited sufficiently within the text.

Report

This paper examines the Q-system approach to the XXZ chain for q a root of unity. In particular it addresses:

1) How to handle the apparently infinite number of solutions of the Q-system resulting from Fabricus-McCoy (FM) strings. 2) After having modified the Q-system in order to remove the FM strings (and roots at infinity), how to add them back to obtain a complete set of Bethe roots.

Point 1) is achieved, at least in simple examples, by looking at the relationships between the $c_i$ coefficients that occur for FM strings (and also roots at infinity) and eliminating solutions that satisfy these relationships by adding a corresponding additional constraint into the Q-system. This is an effective, if brutal, technique, but it does of course require knowledge of the $c_i$ relations. There does not seem to be a general expression for these relations beyond the simple examples given, which is understandable but a limitation on the general applicability of the method.

Point 2) seems to me to be the key result of the paper, as expressed in the 'claim' given by Equaiton (5.3). The claim is that for a given primitive solution of the Q-system, all descendents associated with this solution are given by finding the zeros of the rational function F. If true, then despite the complexity of finding the zeros of this function, this is an important result. The authors state that the claim is proved in Appendix B, but as it stands there is insufficent detail given in Appendix B for me at least to be able to reconstruct the claimed proof. Several simple examples are given that are consistent with the claim.

Overall, this is an interesting paper that looks at extending the Q-system approach to the subtle root of unity case. It is certainly of potential interest to the community, and I believe could be worthy of ultimate publication in SciePost. However, I have the below reservations (expressed in the Requested Changes section of the report) about both the style and content of the manuscript in its current form.

While many of these points are minor (apart from the missing citations and proof), they and other similar examples add up to make the paper unnecessarily hard to read. If thIs can be satisfactorily addressed then I would recommend publication.

Requested changes

Warnings issued while processing user-supplied markup:

  • Inconsistency: plain/Markdown and reStructuredText syntaxes are mixed. Markdown will be used.
    Add "#coerce:reST" or "#coerce:plain" as the first line of your text to force reStructuredText or no markup.
    You may also contact the helpdesk if the formatting is incorrect and you are unable to edit your text.

1) Key references are often missing altogther or the citations are not at the appropriate place in the text. For example: 1.1) Section 4.1 - there are no references at all for Q-systems. 1.2) Page 9 paragraph 2 about FM strings and infinity pairs - there should be citations at this point in the text. 1.3) There are no references in the background Section 2.1 apart from [12]. -- 2) The presentation style is often lacking in enough information or citations to be useful: for example 2.1) 'For special situations [...] the Hilbert space has the same structure as the generic q case.' What special situations and where is this discussed in the literature? More generally, the word special is overused in the text to avoid clear explanation. 2.3) 'For example, the scattering phase between an FM string [...] is trivial'. Why is this and where in the literature is it discussed? -- 3) The mathematical presentation is sometimes confusing and at times a little sloppy. For example: 3.1) 'The QQ-relations is defined up to proportionality'. As (4.1) is your definition, this is not the case. 3.2) After the above, Equation (4.2) has a proportionality sign on the rhs. Why not make this equal in order to have a well-defined system solution (as in the existing uncited literature), or is this the proportionality you were talking about? 3.3) Define $t_j^{FM}$ in equation (4.8). I can guess what it should be but I shouldn't have to. 3.4) Beneath (4.20) the argument is very confusing. You define c_i in terms of t_i by (4.20) and then, to paraphrase, you say 'we can eliminate t_j from (4.20'. The only way I could figure out what you meant is by looking at the examples. Also, you intruduce an integer N_c in (4.21) without comment. Do you know what this is on not? Please say either way. 3.5) (4.27): Please comment on what the symbol $\omega$ represents. 3.6) Page 21: The 'Hasse diagram' section doesn't tell me either what a Hasse diagram is or how it relates to the structure of descendents. Again I had to jump forward to example to figure out what you meant. Please either cross reference to the coming explanation, or postpone the discussion until you give details. 3.7) As mentioned above, you need to decide whether your key claim given by Equation (5.3) is a conjecture or theorem. If it is a conjecture backed by examples, say so. If it is a theorem, give precise statment and a clear proof with enough details that a conscientious reader can reconstruct it.

---

## Round 3 · Referee Report · Anonymous · 2024-4-16

Report

As the authors took in to account all of my suggestions and changed the paper accordingly, I support its publication now.

Recommendation

Publish (meets expectations and criteria for this Journal)

---

## Round 3 · Referee Report · Anonymous · 2024-4-17

Report

I am satisfied that the author's have addressed the points I raised in my earlier report, and I am happy to now recommend publication.

Recommendation

Publish (easily meets expectations and criteria for this Journal; among top 50%)

---

## Round 3 · Author Response

Reply to Anonymous Report 1 on 2024-2-3 (Invited Report)

Referee: 1. In the $\ell_2$-dimensional representations of the transfer matrix it is not clear how the representation depends on the complex spin $s$. Is it related to $\ell_2$? Or is it related to the highest weight as for the spin s representations?

Answer: We would like to thank the referee for the question. Complex spin $s$ is not related to $\ell_2$ in general. It can take any complex value. However, when $s = \frac{\ell_2 - 1}{2}$, the representation coincides with the highest weight representation of (half-integer) spin $s = \frac{\ell_2 - 1}{2}$.

Referee: 2. The $v$ and $v^\prime$ notation is not systematic in the paper. In (2.18) did the author mean $v$ or $v^\prime$, what is the definition of $t^\prime$? Also consider (5.1).

Answer: We would like to thank the referee for spotting the typos. We fixed the typos in Eq. (2.18) and (5.1). $t^\prime_m = \exp (v^\prime_m)$, which is also added to the main text.

Referee: 3. In (3.7) and (3.8) $u_m$ should be sent to the infinities and not $t_m$-s.

Answer: We would like to thank the referee for spotting the typos. We fixed the typos in Eq. (3.7) and (3.8).

Referee: 4. In (3.15) and (3.17) $N$ denotes, what was denoted by $M$ before.

Answer: We would like to thank the referee for spotting the typos. We have fixed the typos throughout Sec. 3.2.

Referee: 5. Some articles are not used correctly.

Answer: We would like to thank the referee for the suggestions. We try our best to fix some of the articles in the main text.

Referee: 6. Somehow one exception was Example 5.4 for me. I think the logic could be improved a bit there.

Answer: We would like to thank the referee for the comment. We added more explanations on the mirroring structure of the descendant tower in Example 5.4.

Reply to Anonymous Report 2 on 2024-2-7 (Invited Report)

Referee: 1. Key references are often missing altogether or the citations are not at the appropriate place in the text.

Answer: We would like to thank the referee for the suggestions.

Referee: 1.1) Section 4.1 - there are no references at all for Q-systems.

Answer: We cited several papers on rational Q-systems in the introduction section. We cite them again in Sec. 4.

Referee: 1.2) Page 9 paragraph 2 about FM strings and infinity pairs - there should be citations at this point in the text.

Answer: We cited papers on FM-strings and infinity pairs in the introduction section. We cite them again on page 9.

Referee: 1.3) There are no references in the background Section 2.1 apart from [12].

Answer: We add more citations on the integrability of the XXZ spin chain in Sec. 2.1.

Referee: 2. The presentation style is often lacking in enough information or citations to be useful: for example
2.1) 'For special situations [...] the Hilbert space has the same structure as the generic q case.' What special situations and where is this discussed in the literature? More generally, the word special is overused in the text to avoid clear explanation.

Answer: We would like to thank the referee for the comments. We changed the sentence to "For a generic twist $\kappa$", and reduced the usage of the word "special" in the main text.

Referee: 2.3) 'For example, the scattering phase between an FM string [...] is trivial'. Why is this and where in the literature is it discussed?

Answer: The scattering phase between any FM string and other Bethe roots is trivial because Eq. (3.10), the scattering phase $\prod_{n=1}^{\ell_2} S(u, u_n) = 1$. This is a straightforward exercise, and it is well-known in the literature. We added a few citations here.

Referee: 3. The mathematical presentation is sometimes confusing and at times a little sloppy. For example:
3.1) 'The QQ-relations is defined up to proportionality'. As (4.1) is your definition, this is not the case.
3.2) After the above, Equation (4.2) has a proportionality sign on the rhs. Why not make this equal in order to have a well-defined system solution (as in the existing uncited literature), or is this the proportionality you were talking about?

Answer: We would like to thank the referee for the comments. We would like to remark here that the proportionality when defining the QQ-relations is quite common in the previous literature. Nevertheless, we deleted the sentence "The QQ-relations is defined up to proportionality", and fixed the notation such that all the formulae are defined as equalities.

Referee: 3.3) Define $t^{\rm FM}_j$ in equation (4.8). I can guess what it should be but I shouldn't have to.

Answer: We would like to thank the referee for the comment. We could not find $t^{\rm FM}_j$ in Eq. (4.8), but we presumed that the referee was referring to Eq. (4.18). We now added the definition of $t^{\rm FM}_j$ as the exponential of the FM-string Bethe roots in Eq. (3.9).

Referee: 3.4) Beneath (4.20) the argument is very confusing. You define $c_i$ in terms of $t_i$ by (4.20) and then, to paraphrase, you say `we can eliminate $t_j$ from (4.20)'. The only way I could figure out what you meant is by looking at the examples. Also, you introduce an integer $N_c$ in (4.21) without comment. Do you know what this is on not? Please say either way.

Answer: We would like to thank the referee for the comment. We added a sentence below Eq. (4.20) to explain the relation between $t_j$ and $c_j$ from the elimination theory. We added an explanation on the meaning of $N_c$ below Eq. (4.21).

Referee: 3.5) (4.27): Please comment on what the symbol $\omega$ represents.

Answer: We would like to thank the referee for the comment. In Eq. (4.27) we used the variable $w$ to ensure the constraint $\left(|R_1|^2+\ldots+|R_{N_c}|^2\right)$ remains non-zero. We added the comment below Eq. (4.27).

Referee: 3.6) Page 21: The `Hasse diagram' section doesn't tell me either what a Hasse diagram is or how it relates to the structure of descendants. Again I had to jump forward to example to figure out what you meant. Please either cross reference to the coming explanation, or postpone the discussion until you give details.

Answer: We would like to thank the referee for the comment. We added a citation of the definition of the Hasse diagram from Wikipedia.

Referee: 3.7) As mentioned above, you need to decide whether your key claim given by Equation (5.3) is a conjecture or theorem. If it is a conjecture backed by examples, say so. If it is a theorem, give precise statement and a clear proof with enough details that a conscientious reader can reconstruct it.

Answer: We would like to thank the referee for the comment. We confirm that Eq. (5.3) is indeed a theorem. We added the details of the derivation below (B.8) on the structure of the zeros of $P(u)$ and how to obtain (5.3). We hope that now we elucidate the method to obtain the zeros of FM-strings in the main text.

---

## Round 3 · List of Changes

1. We fixed typos in the previous version of the draft. For example, in Eqs. (2.18), (3.7), (3.8), (3.15), (3.17) and (5.1).

2. We added citations in Sec. 2 and 4 according to the referee's suggestions.

3. We the definition of $t^{\rm FM}_j$ in Eq. (4.18).

4. We added the comment below Eq. (4.27) on the additional constraints in the Q-system.

5. We changed several equations in Sec. 4 to remove the ambiguity that "The QQ-relations is defined up to proportionality" (which sentence has been removed from the article too).

6. We added further explanation in Example 5.4.

7. We added the details in Appendix B to explain the derivation of Eq. (5.3).

8. Small changes regarding the misused articles has been performed.

---

## Editorial Decision

accepted_in_target_journal